# Minimally Invasive Machine Unlearning via Posterior Control

## Abstract

Approximate Machine Unlearning (MU) methods typically forget specific data by modifying model parameters or learning data-dependent augmentations. However, each unlearning request requires a dedicated optimization process, resulting in high computational overhead and cumulative degradation of model performance over time. Recent approaches have proposed manipulating a small subset of neural activations as a more targeted alternative, yet these methods still rely on on-demand searches for relevant parameters and remain computationally expensive. We propose Minimally Invasive Machine Unlearning (MIMU), a posterior control–based MU framework that enables unlearning without inference-time optimization. MIMU introduces a parameterized control module that maps unlearning requests to binary masking policies via forward propagation. The resulting masks identify the most influential parameters associated with the forget set, enabling effective predictive ability suppression without additional optimization, when an unlearning request is issued. Experimental results across multiple benchmarks demonstrate that MIMU achieves competitive unlearning performance while preserving the generalization ability and prediction patterns of the original model, offering a favorable trade-off between unlearning efficiency and effectiveness.

## 1 Introduction

Many advanced Machine Unlearning (MU) methods adopt approximate unlearning objectives (Nguyen et al., 2025) that update the model by jointly minimizing changes in predictive error on the retained training data while maximizing the error on the data to be forgotten. Although these approaches can effectively erase memorization of target data, they often (1) unintentionally alter the model's overall behavior (Cooper et al., 2025) and (2) require inference-time optimization upon receiving an unlearning request (Wang et al., 2024). As a result, they can degrade the model's generalization ability and substantially increase the computational cost of frequent unlearning operations.

To mitigate the performance degradation caused by frequent unlearning operations, recent research has shifted focus toward methods that operate on a limited number of neural parameters or activations. These techniques aim to reduce the computational footprint and minimize disruption to the model's learned representation by identifying and updating only the weights most relevant to the data points to be forgotten. For instance, Mehta et al. (2022) uses conditional independence analysis to identify semantically relevant parameters for each data point. Similarly, Malekmohammadi et al. (2025) proposes selecting the sharpest parameters, those with the largest Hessian diagonal values, for unlearning updates. Other approaches, such as Torkzadehmahani et al. (2024), reset and fine-tune only the most critical parameters identified through memorization analysis. Layer-specific methods like Gogineni & Nadimi (2024) combine pruning and update tracking to localize unlearning to particular layers, while Cha et al. (2023) employ adversarial relabeling and weight importance to guide parameter changes. More recently, Li et al. (2025) introduced a patching framework with certifiable guarantees that modifies localized parameter regions. These works show that MU can be achieved through targeted parameter intervention, though they often suffer from high computational complexity due to the need of instance-specific parameter search or fine-tuning.

We propose Minimally Invasive Machine Unlearning (MIMU), the first posterior control–based machine unlearning framework that eliminates inference-time optimization for individual unlearning requests, thereby significantly reducing both computational cost and model invasiveness. Inspired by posterior control theory

(Li & Rush, 2020), MIMU learns a parameterized probabilistic control module that maps an unlearning request (expressed as a forget set) to a sparse parameter mask identifying the parameters most responsible for the target predictions. Unlike iterative, instance-specific optimization methods (Deo et al., 2025) or layer-wise heuristic approaches (Xu et al., 2024; Dong et al., 2017), MIMU trains and caches its control module prior to receiving any unlearning request, enabling highly efficient processing at inference time. Extensive experiments demonstrate that a well-trained MIMU control module reliably identifies high-influence parameter subsets for arbitrary unlearning requests. By directly masking these parameters, MIMU effectively removes predictive capability on the forget set while preserving both predictive performance and decision patterns on retained data, achieving competitive results against state-of-the-art unlearning baselines. Further analysis shows that, in addition to its exceptional inference-time efficiency, MIMU exhibits several desirable properties: (1) it preserves prediction pattern of model on retain data points (evidenced by saliency map analysis), and (2) it drives predictive confidence on the forget set toward high predictive entropy, rather than confidently misclassifying forgotten samples into incorrect classes that are observed in many existing methods.

## 2 Preliminary

MU methods are divided into two classifications, exact and approximate unlearning (Neel et al., 2021; Thudi et al., 2022b). Exact unlearning requires training the model such that retraining it without the forget data is more efficient than naively retraining the entire model from scratch. Early works on exact unlearning achieved this by exploiting additive (summation-based) training formulations (Cao & Yang, 2015), or smart division of the data for targeted unlearning (Bourtoule et al., 2020; Yan et al., 2022a). While standard exact machine unlearning (MU) methods are more efficient than full retraining, they can still suffer substantial computational costs or high storage overhead from maintaining and ensembling models trained on disjoint data shards (Yan et al., 2022b). Approximate unlearning is a faster and more cost-efficient approach for reducing the predictive confidence of the model on the specific set of points. Indeed approximate unlearning methods (Guo et al., 2023; Golatkar et al., 2020; Thudi et al., 2022a; Warnecke et al., 2023; Kurmanji et al., 2023; Li et al., 2024) succeed in time and storage efficiency but they require updates across all parameters within the model which can lead to deterioration of the model's generalization.

To preserve the computation efficiency and reduce damage on the unlearned model's generalization, recent developments (Wu et al., 2022; Jia et al., 2024) have focused on targeting the parameters of the model that correspond to the data points to be unlearned. Mehta et al. (2022) perform MU on particular rows (or "slices") of layers within a neural network instead of the entire layer such that an expensive inverse Hessian adjustment can be performed piece by piece. Goel et al. (2023) "freezes" all but the last k layers of a DNN and offer two approaches, retraining those k layers from scratch or fine-tuning the k layers on the retain dataset. Focusing on the later layers allows the model to keep all the lower-level representations it has learned in hopes of preserving the model's generalization. Cha et al. (2023) measure a weight's importance with "memory aware synapses" (Aljundi et al., 2018) and use that weight importance to penalize changes to weights that are not important to the forget set during finetuning. Malekmohammadi et al. (2025) adopt a layer-wise approach by computing the average Hessian for each layer and performing unlearning on the layer with the flattest curvature. The authors argue that layers associated with flatter loss landscapes are more difficult to unlearn; hence, successful unlearning in such layers is expected to be more resistant to relearning attacks. Fan et al. (2024) examined the saliency of model weights with respect to the forget set and mask weights whose gradients exceed a predefined threshold. Such weights are deemed more influential for the data to be unlearned and are therefore removed during the unlearning process. Ding et al. (2025) refined this approach to check for crossover importance for both the retain and forget dataset, advising to abstain from masking any neurons with too much importance across both sets. Torkzadehmahani et al. (2024) take an adapted approach to saliency and determine the most important weights for the forget set by the product of the magnitude of the weight and the gradient of the loss for the forget data set.

## 2.1 Objectives of Machine Unlearning

While most machine unlearning (MU) methods are motivated by privacy concerns, the community has converged on two closely related but distinct interpretations of MU. The original MU definition requires the unlearned model to behave indistinguishably from a model retrained from scratch (Bourtoule et al., 2021), primarily to satisfy Right-to-be-Forgotten regulations found in legislation such as the General Data Protection Regulation (GDPR) in the European Union. However, evaluating unlearning effectiveness by direct comparison with a retrained model remains contentious (Wichert & Sikdar, 2024). In contrast, a more operational but implicit definition emphasizes explicitly suppressing predictive capability on the forget set while preserving generalization on the retain data (Wu et al., 2022). This objective better aligns with the need of repairing deployed models (by suppressing predictive pattern of data points that encode undesirable properties). Notably, although many existing methods are introduced under the first definition, their training objectives and loss formulations more closely reflect the second interpretation.

Our work aligns more closely with this second interpretation: preserving the model's predictive performance on the retain data and maximizing the degradation of the model's accuracy and confidence on the forget set. In particular, rather than merely encouraging misclassification on the forget set, we also aim to reduce the model's predictive confidence on the forget examples by maximizing prediction entropy, thereby making the unlearned model more robust to membership inference attacks (Shokri et al., 2017). Nevertheless, in our experiments, we still include comparisons against fully retrained models, as is common practice in prior work (Chundawat et al., 2022). Here, the minimal invasion emphasizes achieving unlearning with minimal perturbation of prediction patterns on the retain dataset to preserve overall model behavior as first priority and then effectively forgetting the target data as much as possible.

## 3 Minimally Invasive Machine Unlearning

Now, we present Minimally Invasive Machine Unlearning (MIMU), a control-based MU model framework to facilitate forget set predictability removal through dynamic weight masking. Given a trained predictive model $\hat{Y} = f_\theta(X)$ and a set of training data samples $D_{\text{train}} = \{(x, y)\}_m$, we aim to learn a masking function $Q_\phi(\cdot)$ that upon receiving an unlearning request defined by a forget set, produces a masking policy $\mathbf{m}_{\text{forget}} \in \mathbb{B}^{|\theta|}$. This policy identifies the subset of parameters in the predictive model that is most responsible for supporting prediction of forget set. In other words, the mask generated should maximize the alignment between the original model's predictions in the forget set before and after applying the mask, so that:

$$|P_\theta(y_i|x_i) - P_{\mathbf{m}_{\text{forget}} \odot \theta}(y_i|x_i)| \leq \epsilon \quad \forall (x_i, y_i) \in D_{\text{forget}} \tag{1}$$

for some arbitrary small $\epsilon$ and satisfies the sparsity condition $||\mathbf{m}_{\text{forget}}||_0 = \delta$, with pre-defined sparsity target $\delta$. Here, $\odot$ denotes element-wise multiplication. By masking out (disabling) the identified parameters with $1 - \mathbf{m}_{\text{forget}} \odot \theta$, one can remove the predictability of the model on the forget set.

Departing from existing unlearning paradigms that update model parameters or optimize additional parameters for each unlearning request, MIMU does not directly alter the original predictive model. Instead, it treats each unlearning request as a model patching operation that adjusts model behavior through masking out small group of parameters.

### 3.1 Posterior Control for Optimal Mask Searching

The core idea of performing machine unlearning via a learned masking function naturally aligns with the objective of model control, in which a control policy is designed to steer a system toward a desired reference state (here, the removal of predictability on the forget set) while respecting resource and operational constraints. In our case, the masking function is a control module.

If we formalize the control module in a probabilistic setting, where the masking policy produced by the controller is modeled as a stochastic policy drawn from a parametric distribution, we can learn a parameterized control module by reducing KL-divergence with respect to an ideal but intractable masking policy $P$ (or

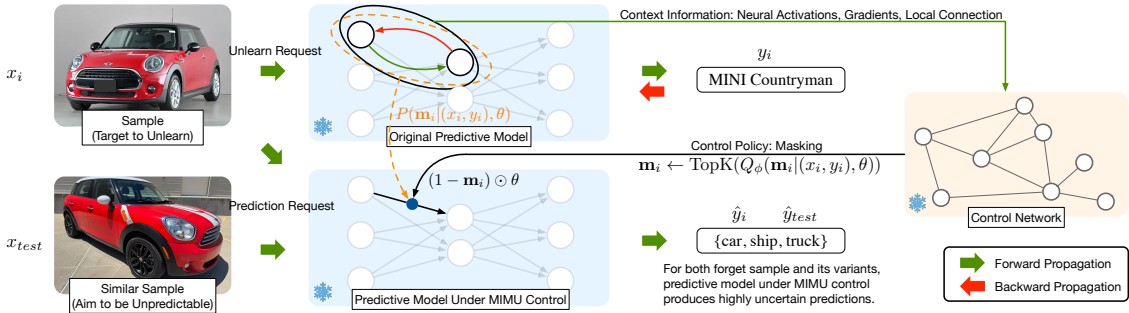

Figure 1: Illustrative figure of MIMU during inference time. Given a pre-trained parameterized control network $Q_\phi(\mathbf{m}_i|(x_i, y_i), \theta)$ that approximate the intractable masking distribution $P(\mathbf{m}_i|(x_i, y_i), \theta)$, the control network will conduct machine unlearning task by applying masked weights upon receiving unlearning request, thereby removing the model's predictive ability regarding similar input data without inference time optimization.

reference distribution).

$$\arg\min_\phi KL\left[Q_\phi(\mathbf{m}_i|(x_i, y_i), \theta)||P(\mathbf{m}_i|(x_i, y_i), \theta)\right]$$
$$s.t. \quad ||\mathbf{m}_i||_0 = \delta, \tag{2}$$

where the reference distribution $P$ is defined as probability of masking given observed predictive model $\theta$ and data $i$

$$P(\mathbf{m}_i|(x_i, y_i), \theta) = \frac{P_{\mathbf{m}_i \odot \theta}(y_i|x_i)\cancel{P(x_i)}P(\mathbf{m}_i|\theta)}{P_\theta(y_i|x_i)\cancel{P(x_i)}}. \tag{3}$$

As a side note, some existing approaches (Labach & Valaee, 2020; Nguyen et al., 2022; Salehinejad & Valaee, 2022), directly maximize the intractable distribution P with respect to $\mathbf{m}_i$ for each data point $i$ as the Maximize a Posterior (MAP) objective. Unfortunately, the optimization has to be conducted through MCMC-like algorithms (Chen et al., 2014; Ma et al., 2015) that iteratively search for the best mask, leading to significant computational cost.

By transforming Equation 2 with respect to the decomposition of the reference distribution (Equation 3), we instead look for modeling $\mathbf{m}_i$ as an parameterized distribution through the following training objective

$$\arg\min_\phi \max_\lambda -E_{Q_\phi}[\log P_{\mathbf{m}_i \odot \theta}(y_i|x_i)]+$$
$$KL[Q_\phi(\mathbf{m}_i|(x_i, y_i), \theta)||P(\mathbf{m}_i|\theta)] + \lambda(||\mathbf{m}_i||_0 - \delta), \tag{4}$$

which is the combination of a cross entropy term, a KL regularization term, and a Lagrange multiplier term. The prior distribution $P(\mathbf{m}_i|\theta) \coloneqq P(\mathbf{m}|\theta)$ by default is a categorical distribution tied to the predictive model's parameter distribution. In particular, given the conclusion from previous study Jia et al. (2024), we set the prior distribution as

$$P(\mathbf{m}|\theta) = \frac{1}{\sum_{k'} ||\theta_{k'}||}||\theta_k|| \quad \forall k \in \{1 \cdots |\theta|\} \tag{5}$$

to reflect the intuition that larger weights account for larger influence on the model's prediction.

Given that both $Q_\phi(\mathbf{m}_i|(x_i, y_i), \theta)$ and $P(\mathbf{m}|\theta)$ are categorical distributions, the KL regularization term in Equation 4 can be minimized by reducing cross-entropy loss and encouraging diversity

$$KL[Q_\phi(\mathbf{m}_i|(x_i, y_i), \theta)||P(\mathbf{m}|\theta)]$$
$$= -\int_{\mathbf{m}_i} Q_\phi(\mathbf{m}_i|(x_i, y_i), \theta)\log P(\mathbf{m}|\theta)d\mathbf{m}_i$$
$$+ \int_{\mathbf{m}_i} Q_\phi(\mathbf{m}_i|(x_i, y_i), \theta)\log Q_\phi(\mathbf{m}_i|(x_i, y_i), \theta)d\mathbf{m}_i. \tag{6}$$

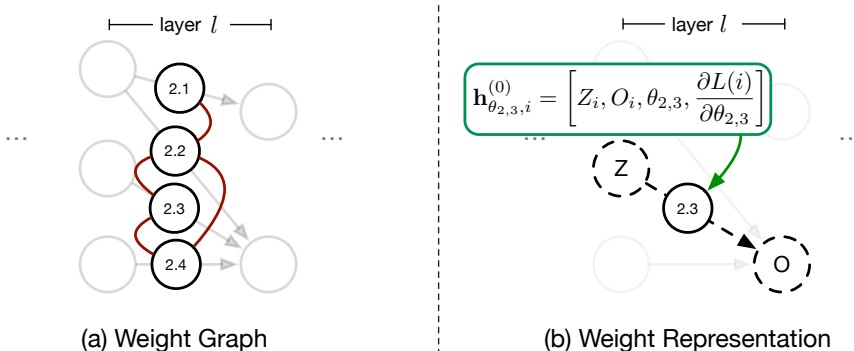

(a) Weight Graph | (b) Weight Representation

Figure 2: Illustrative figure to show the parameter graph construction on a layer of neural network. (a) a pair of parameters $\theta_k$ and $\theta_{k'}$ will have a connection, when $\theta_k \rightarrow v \wedge \theta_{k'} \rightarrow v$ or $v \rightarrow \theta_k \wedge v \rightarrow \theta_{k'}$ with shared node $v$. (b) the feature of each parameter $\theta_k$ consists of the surrounding information.

Optimizing Equation 4 is still hard due to the difficulty of enforcing the duality of the Lagrange multiplier term. Although implementing L1 regularization can effectively approximate the objective function, it often violates the sparsity constraint. To avoid optimization duality, we approximate the objective function by top-$\delta$ sampling during training so that

$$\mathbf{m}_i \sim \text{Top-}\delta(Q_\phi(\mathbf{m}_i|(x_i, y_i), \theta)), \tag{7}$$

where the number of positive samples ($m_{i,k} = 1$) can be enforced through $Q_\phi(m_{i,k}|(x_i, y_i), \theta)$ top-$\delta$ filtering. With the controlled sampling, we now summarize the final implementable training objective as

$$\arg\min_\phi \sum_{(x_i, y_i) \sim D} [-E_{\mathbf{m}_i \sim \text{Top-}\delta(Q_\phi)}[\log P_{\mathbf{m}_i \odot \theta}(y_i|x_i)]$$
$$- E_{\mathbf{m}_i \sim \text{Top-}\delta(Q_\phi)}[\log P(\mathbf{m}|\theta)] + E_{\mathbf{m}_i \sim \text{Top-}\delta(Q_\phi)}[\log Q_\phi]], \tag{8}$$

where we use $Q_\phi$ to denote $Q_\phi(\mathbf{m}_i|(x_i, y_i), \theta)$ for better clarity. Note that, while the parameter $\phi$ in the proposed distribution $Q$ appears non-differentiable due to passing through a sampling step, we note the Gumbel reparameterization with a Top-K softmax subset filtering (Jang et al., 2017; Maddison et al., 2017; Xie & Ermon, 2021) can effectively smooth the path and make the objective trainable with respect to $Q_\phi$. A full description of our implementation of the Gumbel reparameterization is given in Appendix F

### 3.2 Graph Convolution based Control Module

Modern neural predictive models often contain millions or even billions of parameters, making it impractical to design a control module $Q_\phi(m_{i,k}|(x_i, y_i), \theta)$ that jointly conditions on the input data and the full parameter set while producing a mask over all parameters in a single step. To address this scalability challenge, we restrict the scope of masking to a selected layer and instantiate the control module as a graph convolutional network (GCN).

Given a trained predictive model and a selected layer $l$, we represent the parameters of that layer as a graph $G = (V, E)$. Each node $v \in V$ corresponds to an individual weight parameter $\theta_k$ in the selected layer. An edge $e \in E$ connects a pair of parameters $(\theta_k, \theta_{k'})$ if they are associated with the same neuron in the predictive model, either through incoming or outgoing connections, as shown in Figure 2.

The feature of each node $\theta_k$ with respect to each training data point $(x_i, y_i)$ consists of 1) value of inbound activation $z(x_i)$ of layer $l$, 2) outbound activation $o(x_i)$, 3) weight value $\theta_k$, and 4) derivative of the weight $\frac{\partial L(f_\theta(x_i), y_i)}{\partial \theta_k}$.

GCN estimates node embeddings through message passing such that

$$\mathbf{h}_{\theta_k, i}^{(t)} = \sigma \left( A^{(t)} \cdot \frac{1}{\mathcal{N}(\theta_k)} \sum_{\theta' \in \mathcal{N}(\theta_k)} \mathbf{h}_{\theta_{k'}, i}^{(t-1)} + B^{(t)} \cdot \mathbf{h}_{k, i}^{(t-1)} \right), \tag{9}$$

where $A^{(t)}$ and $B^{(t)}$ are trainable parameters of the graph convolution network and $\mathbf{h}_{\theta_k,i}^{(0)}$ is the input feature $\theta_k$ of data point $i$. The output of each node from the graph convolution network is a probability of masking

$$Q_\phi(m_{i,k}|(x_i,y_i),\theta) = \text{Softmax}\left(W \cdot \mathbf{h}_{\theta_k,i}^{(T)} + b\right), \tag{10}$$

where the parameter group $\phi$ denotes all parameters of the GCN such that $\phi = \{\{A^{(t)}, B^{(t)}\}_{t\in\{1:T\}} \cup W\}$.

Although using a graph convolution-based masking function helps address local connectivity concerns, it can still struggle when scaling to model a layer with millions of parameters. Fortunately, prior work (Fel et al., 2024; Dorszewski et al., 2025) has shown that the early layers of neural networks primarily act as generic feature extractors and contribute less directly to specific predictions. This insight allows us to reduce computational overhead by focusing the masking and unlearning process on the last one to two layers of the network, where they have the most direct influence on predictions but with far less number of parameters.

### 3.3 Unlearning via Filtered Posterior Control

Given the learned control module that obtains the important parameter subset for each training data point, we can process unlearning request for one specific training sample $(x_i, y_i)$ by masking the minimal subset of model parameters that uniquely support its/their prediction.

While directly masking all top-$\delta$ parameters from control module as described in Equation 7 may allow us to reduce the predictability of forget data point $(x_i, y_i)$, it may also harm the model's generalization by disrupting predictive patterns shared by some retain data points.

To reduce the risk of overshooting, we filter the masking suggestions from the control module by selecting a representative sample and applying a binary masking operation:

$$\tilde{\theta} \leftarrow \theta \odot (1 - \text{filter}(\mathbf{m}_i, \kappa)),$$

where $\text{filter}(\cdot)$ denotes weighted filtering function that selects top $\kappa \ll \delta$ parameters to mask. In particular, we may stop masking when 1) the predictive confidence $f_{\tilde{\theta}}(x_i)[y_i]$ of forget sample drops below a predictive confidence threshold (meaning successful unlearning), 2) the predictive confidence of retain samples drops below the performance degradation tolerance, or 3) let $\kappa$ to be a percentage of $\delta$. In other words, $\kappa$ can be dynamically selected for better unlearning outcome. In our experiments later, we enforce $\delta \leq 10\%$ of parameters in the selected layer of the target predictive model.

The above described unlearning strategy can be easily expand to set unlearning, where one may want to unlearn multiple training data at once. The only modification is to replace individual masking proposal $\mathbf{m}_i$ from the control module with an aggregated union of masks from all $N$ forget set samples such that

$$\mathbf{m}_{\text{forget-set}} = \bigvee_{i=1}^{N} \mathbf{m}_i \tag{11}$$

In practice, when training samples in the forget set share common predictive patterns, we can subsample from the $N$ samples to reduce computational cost. We also select a smaller $\kappa$ so that $\kappa \cdot N \ll \delta$

## 4 Experiments and Evaluations

**Datasets and models** We evaluate the proposed machine unlearning method, MIMU, against several unlearning algorithms on standard image classification benchmarks, including MNIST, Fashion-MNIST (FM-NIST), CIFAR-10, and ImageNet100. For MNIST, we employ a multilayer perceptron (MLP) as the predictive model. For FMNIST, we use a convolutional neural network (CNN). Experiments on CIFAR-10 are conducted using the ResNet-18 architecture, while for ImageNet100 we adopt ResNet-50 (Appendix B).

**Baselines** We compare the proposed approach against a set of machine unlearning baselines, using *Retraining from scratch* (Retrained) as a reference. The baselines include Random Labelling (RL) (Li & Ghosh, 2023),

Table 1: Class-level unlearning performance across candidate algorithms in terms of retain and forget set accuracy. Results are averaged over 5 runs with standard deviation error bars.

| Method | MNIST | | FMNIST | | CIFAR10 | | ImageNet100 | |
|---|---|---|---|---|---|---|---|---|
| | Forget | Retain | Forget | Retain | Forget | Retain | Forget | Retain |
| Original | $100.00 \pm 0.00$ | $100.00 \pm 0.00$ | $89.12 \pm 0.00$ | $93.39 \pm 0.00$ | $99.96 \pm 0.00$ | $99.87 \pm 0.00$ | $98.54 \pm 0.00$ | $92.57 \pm 0.00$ |
| Retrained | $0.00 \pm 0.00$ | $100.00 \pm 0.00$ | $0.00 \pm 0.00$ | $99.37 \pm 0.52$ | $0.00 \pm 0.00$ | $100.00 \pm 0.00$ | $0.00 \pm 0.00$ | $92.14 \pm 0.04$ |
| Random Labels | $17.79 \pm 10.49$ | $72.86 \pm 6.15$ | $54.24 \pm 4.23$ | $91.08 \pm 0.53$ | $2.55 \pm 4.36$ | $15.16 \pm 09.65$ | $95.58 \pm 02.84$ | $80.34 \pm 01.86$ |
| Catastrophic Forgetting | $99.79 \pm 0.11$ | $100.00 \pm 0.00$ | $30.24 \pm 10.94$ | $94.29 \pm 0.34$ | $0.11 \pm 0.15$ | $99.17 \pm 0.41$ | $13.32 \pm 20.19$ | $89.56 \pm 0.55$ |
| Exact Unlearning | $0.00 \pm 0.00$ | $98.91 \pm 0.53$ | $0.00 \pm 0.00$ | $81.65 \pm 2.16$ | $0.00 \pm 0.00$ | $99.56 \pm 0.09$ | $8.54 \pm 8.29$ | $90.66 \pm 0.24$ |
| NegGrad+ | $0.00 \pm 0.00$ | $95.77 \pm 0.35$ | $0.00 \pm 0.00$ | $79.72 \pm 0.44$ | $0.00 \pm 0.00$ | $99.45 \pm 0.14$ | $0.00 \pm 0.00$ | $87.84 \pm 0.40$ |
| L1-Sparse | $0.00 \pm 0.00$ | $99.77 \pm 0.01$ | $0.00 \pm 0.00$ | $88.48 \pm 0.21$ | $0.00 \pm 0.00$ | $99.80 \pm 0.02$ | $0.08 \pm 0.00$ | $90.47 \pm 0.03$ |
| Saliency Unlearning | $0.18 \pm 0.37$ | $100.00 \pm 0.00$ | $0.00 \pm 0.00$ | $95.62 \pm 0.16$ | $0.00 \pm 0.00$ | $94.63 \pm 0.39$ | $0.00 \pm 0.00$ | $95.31 \pm 0.47$ |
| SSD | $23.24 \pm 6.34$ | $100.00 \pm 0.00$ | $3.90 \pm 1.04$ | $92.36 \pm 0.62$ | $4.03 \pm 7.82$ | $95.33 \pm 1.67$ | $0.00 \pm 0.00$ | $92.55 \pm 0.01$ |
| SCRUB | $0.00 \pm 0.00$ | $98.68 \pm 0.01$ | $0.54 \pm 0.11$ | $92.32 \pm 0.02$ | $0.00 \pm 0.00$ | $99.81 \pm 0.00$ | $0.00 \pm 0.00$ | $90.70 \pm 0.03$ |
| MIMU | $0.78 \pm 1.32$ | $98.78 \pm 0.36$ | $0.73 \pm 0.52$ | $92.55 \pm 0.46$ | $0.00 \pm 0.00$ | $99.64 \pm 0.02$ | $0.00 \pm 0.00$ | $90.72 \pm 0.08$ |

Table 2: Sub-class/random unlearning performance across candidate algorithms in terms of retain and forget set accuracy. Sub-classes are randomly selected from k-means clusters. Results are averaged over 5 runs with standard deviation error bars.

| Method | MNIST | | FMNIST | | CIFAR10 | | ImageNet100 | |
|---|---|---|---|---|---|---|---|---|
| | Forget | Retain | Forget | Retain | Forget | Retain | Forget | Retain |
| Original | $100.00 \pm 0.00$ | $100.00 \pm 0.00$ | $76.10 \pm 0.00$ | $93.29 \pm 0.00$ | $99.88 \pm 0.00$ | $99.88 \pm 0.00$ | $97.58 \pm 0.00$ | $92.62 \pm 0.00$ |
| Retrained | $98.89 \pm 0.11$ | $100.00 \pm 0.00$ | $40.21 \pm 9.87$ | $99.36 \pm 0.69$ | $56.08 \pm 1.95$ | $99.99 \pm 0.03$ | $97.41 \pm 0.38$ | $92.22 \pm 0.03$ |
| Random Labels | $4.97 \pm 0.98$ | $90.40 \pm 0.08$ | $38.30 \pm 6.660$ | $89.13 \pm 0.45$ | $97.36 \pm 2.34$ | $94.38 \pm 4.06$ | $93.39 \pm 5.66$ | $86.31 \pm 1.24$ |
| Catastrophic Forgetting | $98.79 \pm 0.91$ | $98.84 \pm 0.31$ | $68.85 \pm 9.97$ | $92.91 \pm 0.50$ | $88.47 \pm 9.81$ | $99.09 \pm 0.31$ | $96.69 \pm 1.55$ | $89.41 \pm 0.52$ |
| Exact Unlearning | $99.07 \pm 0.46$ | $99.35 \pm 0.09$ | $30.09 \pm 18.80$ | $76.67 \pm 1.70$ | $98.60 \pm 1.05$ | $99.55 \pm 0.11$ | $97.02 \pm 0.46$ | $90.38 \pm 0.52$ |
| NegGrad+ | $96.79 \pm 0.27$ | $99.97 \pm 0.01$ | $14.55 \pm 0.47$ | $91.04 \pm 0.20$ | $61.18 \pm 0.18$ | $99.44 \pm 0.07$ | $47.82 \pm 1.68$ | $89.05 \pm 0.29$ |
| L1-Sparse | $99.13 \pm 0.06$ | $99.17 \pm 0.01$ | $54.81 \pm 0.81$ | $92.35 \pm 0.04$ | $99.31 \pm 0.14$ | $99.85 \pm 0.01$ | $97.18 \pm 0.00$ | $90.56 \pm 0.02$ |
| Saliency Unlearning | $96.67 \pm 2.25$ | $100.00 \pm 0.00$ | $72.13 \pm 21.32$ | $94.09 \pm 0.58$ | $89.08 \pm 3.85$ | $96.87 \pm 0.56$ | $89.70 \pm 5.95$ | $95.23 \pm 0.25$ |
| SSD | $71.16 \pm 15.74$ | $99.15 \pm 0.83$ | $7.88 \pm 7.16$ | $63.11 \pm 29.36$ | $4.03 \pm 7.82$ | $95.33 \pm 1.66$ | $0.00 \pm 0.00$ | $91.80 \pm 0.02$ |
| SCRUB | $98.84 \pm 0.04$ | $98.40 \pm 0.01$ | $61.41 \pm 0.38$ | $90.07 \pm 0.05$ | $99.26 \pm 0.00$ | $99.85 \pm 0.00$ | $95.89 \pm 0.18$ | $90.85 \pm 0.03$ |
| MIMU | $70.52 \pm 7.81$ | $97.24 \pm 0.76$ | $66.76 \pm 20.64$ | $91.90 \pm 0.23$ | $73.74 \pm 3.46$ | $99.41 \pm 0.05$ | $90.00 \pm 0.87$ | $92.06 \pm 0.08$ |

Catastrophic Forgetting (CF@k) (Goel et al., 2023) Exact Unlearning (EU@k) (Goel et al., 2023) , L1-Sparse (Jia et al., 2024), Saliency Unlearning (SalUn) (Fan et al., 2024), Selective Synaptic Dampening (SSD) (Foster et al., 2024), NegGrad+ (Kurmanji et al., 2023), and SCRUB (Kurmanji et al., 2023) (see Appendix 6). In particular, CF and EU are lightweight MU approaches that operate on a selected $l$-th layer of the model. In our experiments, we apply them to the second-to-last layer, consistent with the MIMU setting.

## 4.1 Overall Unlearning Performance

We next present the quantitative performance evaluation of the candidate unlearning algorithms. The evaluation covers multiple aspects: (1) accuracy on the retain and forget sets after unlearning, (2) prediction uncertainty (measured by entropy). It is worth noting that, while we include retraining from scratch as a reference, our goal is not to match the unlearned model to the retrained one. Instead, the objective is to remove the model's ability to predict the forget set while preserving its performance on the retain set.

Table 1 reports the prediction accuracy of unlearned models on the forget and retain datasets. The results show that MIMU achieves competitive unlearning performance compared to existing approaches that need inference-time optimization. In particular, we note MIMU significantly reduces the model's predictive ability on the forget set (with near 0% prediction accuracy) while effectively preserved model's performance on the

Table 3: Class-level unlearning performance in terms of post-unlearning prediction entropy. Results are averaged over 5 runs with standard deviation error bars.

| Method | MNIST | | FMNIST | | CIFAR10 | | ImageNet100 | |
|---|---|---|---|---|---|---|---|---|
| | Forget | Retain | Forget | Retain | Forget | Retain | Forget | Retain |
| Original | $0.00 \pm 0.00$ | $0.00 \pm 0.00$ | $0.33 \pm 0.00$ | $0.19 \pm 0.00$ | $0.00 \pm 0.00$ | $0.01 \pm 0.00$ | $0.08 \pm 0.00$ | $0.28 \pm 0.00$ |
| Retrained | $0.26 \pm 0.03$ | $0.00 \pm 0.00$ | $0.35 \pm 0.10$ | $0.03 \pm 0.01$ | $0.34 \pm 0.01$ | $0.00 \pm 0.00$ | $2.01 \pm 0.11$ | $0.32 \pm 0.00$ |
| Random Labels | $2.13 \pm 0.05$ | $0.80 \pm 0.12$ | $1.22 \pm 0.02$ | $0.37 \pm 0.01$ | $2.27 \pm 0.03$ | $2.25 \pm 0.01$ | $2.06 \pm 0.53$ | $1.72 \pm 0.20$ |
| Catastrophic Forgetting | $0.01 \pm 0.00$ | $0.00 \pm 0.00$ | $0.76 \pm 0.02$ | $0.17 \pm 0.01$ | $0.66 \pm 0.26$ | $0.04 \pm 0.01$ | $1.46 \pm 0.24$ | $0.22 \pm 0.00$ |
| Exact Unlearning | $0.35 \pm 0.11$ | $0.02 \pm 0.00$ | $1.92 \pm 0.06$ | $1.45 \pm 0.06$ | $1.03 \pm 0.31$ | $0.05 \pm 0.00$ | $1.65 \pm 0.09$ | $0.22 \pm 0.00$ |
| NegGrad+ | $0.00 \pm 0.00$ | $0.00 \pm 0.00$ | $0.01 \pm 0.00$ | $0.04 \pm 0.00$ | $0.19 \pm 0.07$ | $0.00 \pm 0.00$ | $0.04 \pm 0.01$ | $0.02 \pm 0.00$ |
| L1-Sparse | $1.00 \pm 0.02$ | $0.07 \pm 0.00$ | $1.61 \pm 0.01$ | $1.11 \pm 0.00$ | $1.20 \pm 0.03$ | $0.06 \pm 0.00$ | $3.10 \pm 0.02$ | $0.79 \pm 0.00$ |
| Saliency Unlearning | $2.21 \pm 0.01$ | $0.01 \pm 0.00$ | $0.62 \pm 0.02$ | $0.15 \pm 0.00$ | $0.40 \pm 0.11$ | $0.21 \pm 0.02$ | $1.21 \pm 0.19$ | $0.18 \pm 0.01$ |
| SSD | $0.55 \pm 0.02$ | $0.00 \pm 0.00$ | $0.62 \pm 0.07$ | $0.22 \pm 0.00$ | $0.81 \pm 0.07$ | $0.06 \pm 0.01$ | $2.33 \pm 0.04$ | $0.28 \pm 0.00$ |
| SCRUB | $0.96 \pm 0.01$ | $0.10 \pm 0.00$ | $1.22 \pm 0.01$ | $0.45 \pm 0.00$ | $0.93 \pm 0.02$ | $0.03 \pm 0.00$ | $3.27 \pm 0.05$ | $0.82 \pm 0.01$ |
| MIMU | $0.86 \pm 0.05$ | $0.03 \pm 0.00$ | $0.69 \pm 0.06$ | $0.25 \pm 0.01$ | $1.62 \pm 0.02$ | $0.02 \pm 0.00$ | $2.87 \pm 0.03$ | $0.37 \pm 0.01$ |

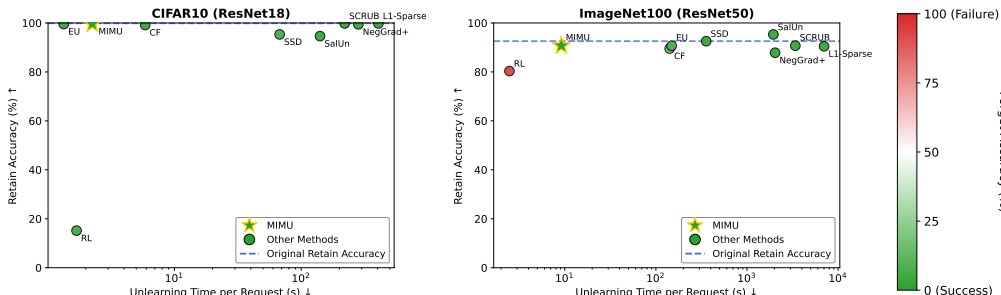

Figure 3: For every unlearning request, MIMU delivers the competitive results quickly. Other common MU techniques deliver similar performance but can cost significantly more in compute time. The graph above shows the Retain Accuracy in percentage on the Y axis, the Unlearning Time measured in wall-clock seconds on the X axis and the Forget Accuracy is measure from the hue of the marker, from red (failure - up to 100% retention) to green (success - down to 0% retention). Time values can be found in Table 9 in the Appendix.

retain dataset (with $\leq 2\%$ performance degradation) over all datasets in the experiments. It is the most stably performed MU algorithm compared to the baseline methods. Further examination of prediction confidence and uncertainty in Table 3 reveals that MIMU substantially increases the entropy of the forget set while maintaining low entropy for the retain set. This demonstrates that MIMU genuinely removes predictive capability rather than merely inducing incorrect predictions with high confidence.

Table 2 presents results for subclass-level unlearning. Subclass-level unlearning is particularly challenging due to strong entanglement between the predictive patterns of the forget set and retain data points within the same class. As a result, many machine unlearning methods struggle to balance effective forgetting with performance preservation on retain data, often leading to two failure modes, either the unlearning process degrades the target model, or it fails to sufficiently weaken predictability on the forget set. Under a fixed performance degradation tolerance, MIMU achieves a trade-off between unlearning effectiveness and retain set performance that leans towards retain set preservation.

## 4.2 Unlearning Efficiency

We next evaluate the computational efficiency of unlearning algorithms by measuring the wall-clock processing time required to handle each unlearning request. Figure 3 summarizes the unlearning effectiveness and computational cost across all methods. The x-axis represents the average unlearning time per request,

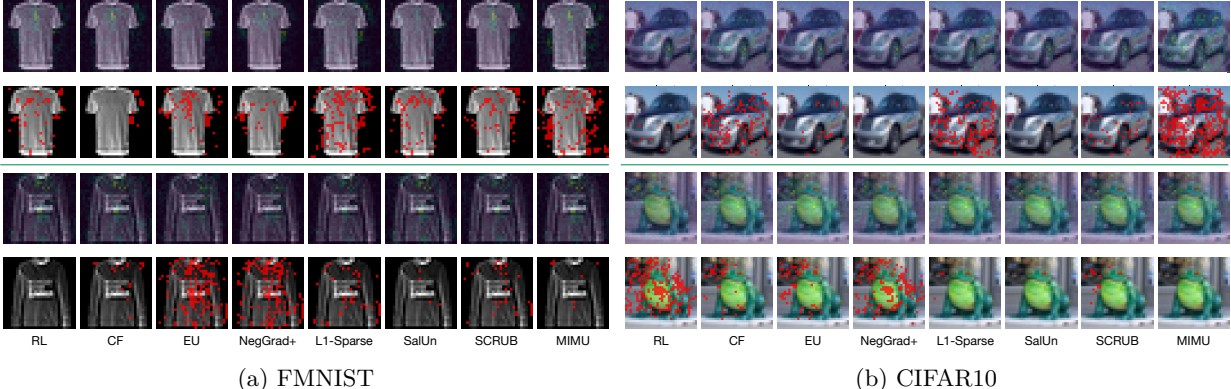

(a) FMNIST  (b) CIFAR10

Figure 4: Saliency map analysis before and after unlearning. The top row shows the saliency map of a forget sample after unlearning. The second row shows the saliency difference between before and after unlearning for the forget sample. The bottom two rows show the corresponding results for a retain sample. Red dots highlight regions with notable saliency differences, which implies prediction pattern shift of target model on the data.

while the y-axis reports the retain accuracy after unlearning. The color gradient indicates the forget accuracy, where green shade corresponds to lower forget accuracy (effective forgetting). The dashed horizontal line denotes the accuracy of the original model before unlearning. MIMU is highlighted with star markers, whereas competing methods are represented by circular markers.

As shown in Figure 3 across both CIFAR-10 and ImageNet-100, MIMU preserves retain accuracy close to that of the original model while simultaneously reducing forget accuracy to zero. Importantly, MIMU demonstrates significantly lower unlearning time per request compared to optimization-based approaches. The computational advantage of MIMU becomes more pronounced as model complexity grows. For example, on ImageNet-100 using a ResNet-50 backbone, methods that require iterative optimization during unlearning experience a substantial increase in runtime, whereas MIMU remains efficient. More detailed runtime comparisons are provided in Table 9. MIMU on CIFAR-10 and ImageNet-100 exhibits a substantial computational advantage over strong baselines , including NegGrad+, L1-Sparse, and SCRUB. While lightweight methods such as CF and EU achieve competitive execution times on smaller datasets, their computational complexity leads to significantly increased runtime when applied to larger models or datasets. Random Labeling (RL) is computationally inexpensive; however, this efficiency comes at the cost of effectiveness. As shown in Tables 1 and 2, RL often degrades generalization performance or fails to meaningfully suppress predictability on the forget set, even when provided with a larger computational budget through extensive hyperparameter tuning. In contrast, although strong baselines such as L1-Sparse and SCRUB achieve unlearning performance comparable to MIMU, they are substantially more computationally demanding, making them less practical for time-sensitive or large-scale unlearning scenarios. MIMU offers a scalable solution that combines strong unlearning performance with low computational overhead. A full description of the components of a MIMU unlearning request is given in Appendix E.

### 4.3 Ablation Study

We conduct an ablation study to analyze the impact of two key hyperparameters in MIMU: (1) the number of representative samples and (2) the top-$\kappa$ masking threshold determined by the top-$\kappa$ weights selected for unlearning. We evaluate these factors on both CIFAR-10 and ImageNet-100 by measuring their effect on the retain-set and forget-set accuracies. For the ablation results on MNIST and FMNIST see Appendix H.

Figure 5 presents the results on ImageNet-100. We observe that MIMU achieves effective unlearning even when using a relatively small number of representative samples and a small top-$\kappa$ threshold. Simultaneously, the method drives the forget-set accuracy close to zero while largely preserving the retain-set accuracy.

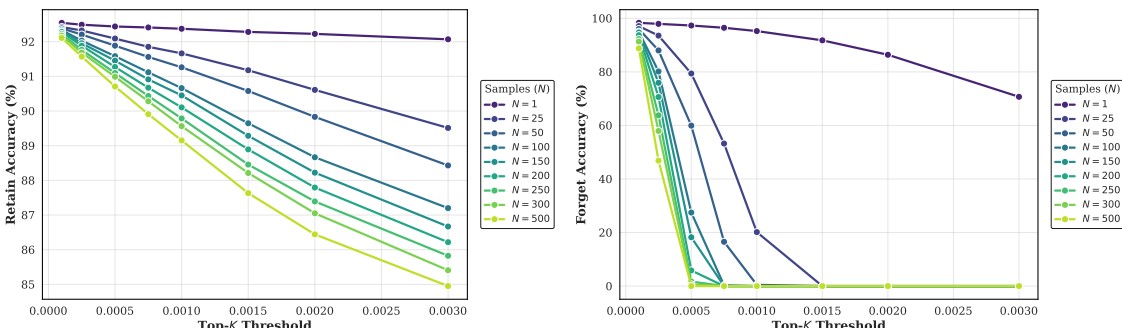

Figure 5: Ablation study of the number of representative samples and top-$\kappa$ masking threshold in MIMU. Results show class-level unlearning performance on ImageNet-100, measured in terms of retain-set and forget-set accuracies after unlearning.

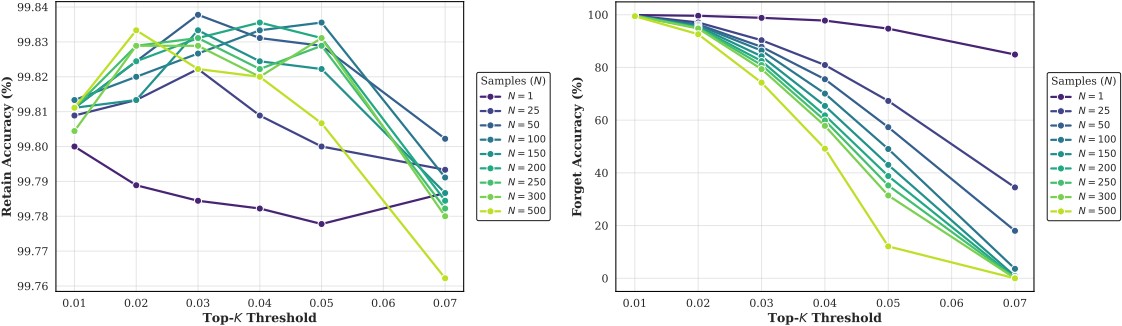

Figure 6: Ablation study of the number of representative samples and top-$\kappa$ masking threshold in MIMU. Results show class-level unlearning performance on CIFAR10, measured in terms of retain-set and forget-set accuracies after unlearning.

As the masking threshold increases, the forget-set accuracy decreases rapidly, indicating that the weights identified by MIMU are highly influential for the target forget classes. This sharp decline suggests that the most impactful forget-related information is concentrated within a relatively small subset of weights. The retain-set accuracy degrades at slower rate that indicates the selected weights primarily encode information associated with the forget set while contributing less to the retained classes.

The results for CIFAR-10 are shown in Figure 6. Similar to ImageNet-100, MIMU causes a substantial reduction in forget-set accuracy while maintaining the retain-set performance. However, compared to ImageNet-100, we employ larger masking thresholds.

### 4.4 Saliency Map Shift after Unlearning

Understanding how prediction patterns change after an unlearning operation is critical for evaluating the safety and reliability of the resulting model. In particular, even when an unlearned model preserves prediction accuracy on retain data, a shift in its internal decision-making process can raise concerns about what features the model relies on to make predictions; whether it begins to depend on superficial or spurious cues (Xiao et al., 2021).

To assess this risk, we conduct a detailed prediction pattern analysis using saliency maps (Simonyan et al., 2014), as shown in Figure 4. For forget set (top two rows), larger saliency difference (more red dots) is better, whereas, for retain set (bottom two rows), smaller saliency difference is better. Our results show that several strong baselines, including RL, EU, L1-Sparse, and NegGrad+, exhibit pronounced prediction pattern shifts on retain data points. This observation is consistent with our earlier findings in Table 1, where these methods demonstrate suboptimal prediction preservation on the retain set.

Interestingly, although SCRUB and MIMU achieve comparable unlearning performance in terms of accuracy, their internal behaviors differ substantially. Specifically, SCRUB induces a much stronger prediction pattern shift on retain samples than MIMU. This result aligns well with our goal of minimally invasive unlearning, as MIMU preserves more consistent model interpretations and avoids dramatic changes in the features used for prediction. More observations are listed in Appendix G.

## 5 Conclusion

In this paper, we introduced MIMU, a model control–based framework for machine unlearning tasks. Unlike existing paradigms that modify model parameters, MIMU takes a different route by controlling a model's predictive behavior through a pre-trained control unit. This design leads to competitive unlearning performance while providing substantial efficiency gains. Our experiments further show that MIMU consistently suppresses the predictability of forget sets. Together, these results highlight model control as a promising new direction for sustainable machine unlearning.

## Impact Statement

We propose a machine unlearning method based on posterior control that efficiently responds to near–real-time unlearning requests without incurring significant computational overhead, enabling an inference-time–optimization–free unlearning paradigm. This approach is particularly well suited for online services that require rapid intervention on deployed machine learning models to mitigate undesired behaviors caused by bad training samples while avoiding substantial performance degradation.

The field of Machine Unlearning offers many useful techniques for ensuring safer and more reliable machine learning models. These techniques can also potentially be used unethically by bad actors who wish to censor or suppress information within these models for malicious or selfish reasons.

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

Table 4: Summary of datasets used in this paper. For each dataset, the corresponding model architecture is specified. The internal dimensions of the MLP models are listed in the table, while for ResNet we follow the original architecture with one additional hidden layer inserted before the classification layer, whose dimension is also reported. For ImageNet-100, the model was initialized with weights trained on ImageNet.

| CIFAR-10 | Benchmark | Image | 60,000 | $32 \times 32 \times 3$ | 10 | No | Yes | ResNet18 (512, 128) |
|---|---|---|---|---|---|---|---|---|
| MNIST | Benchmark | Image | 60,000 | $28 \times 28 \times 1$ | 10 | No | Yes | MLP (128, 64) |
| Fashion-MNIST | Benchmark | Image | 70,000 | $28 \times 28 \times 1$ | 10 | No | Yes | CNN (128, 64) |
| ImageNet-100 | Benchmark | Image | 130,000 | $224 \times 224 \times 3$ | 100 | No | Yes | ResNet50(2048, 256) |

## A  Additional Derivations

Here, we provide additional derivations of the equations in the main paper. All notations, datasets and experimental settings follow the definition of the main paper unless stated otherwise.

### A.1  Equation 3

Following the Bayes rules', the posterior distribution of masking $m_i$ can be derived as product of prior and likelihood terms

$$P(\mathbf{m}_i|(x_i, y_i), \theta) = \frac{P_\theta(x_i, y_i|\mathbf{m}_i) \, P(\mathbf{m}_i|\theta)}{P_\theta(x_i, y_i)}$$

We can re-write the likelihood term with explicit notion where we apply mask $m_i$ to the model parameter $\theta$

$$P_\theta(x_i, y_i|\mathbf{m}_i) := P_{\mathbf{m}_i \odot \theta}(x_i, y_i)$$

By the properties of joint probability:

$$P_{\mathbf{m}_i \odot \theta}(x_i, y_i) = P_{\mathbf{m}_i \odot \theta}(y_i|x_i) \, P(x_i)$$

$$P_\theta(x_i, y_i) = P_\theta(y_i|x_i) \, P(x_i)$$

Thus, we have:

$$P(\mathbf{m}_i|(x_i, y_i), \theta) = \frac{P_{\mathbf{m}_i \odot \theta}(y_i|x_i) \, P(x_i) \, P(\mathbf{m}_i|\theta)}{P_\theta(y_i|x_i) \, P(x_i)}$$

which corresponds to Equation (3) in the main paper.

### A.2  Equation 4

Starting with the motivating objective in Equation 2, we have

$$\begin{aligned} &\mathrm{argmin}_\phi KL\left[Q_\phi(\mathbf{m}_i|(x_i, y_i), \theta)||P(\mathbf{m}_i|(x_i, y_i), \theta)\right] \\ &s.t. \quad ||\mathbf{m}_i||_0 \le \delta, \end{aligned} \tag{12}$$

which, by extend, is defined as following

$$\mathrm{KL}(Q\|P) = E_{Q_\phi}\left[\log \frac{Q_\phi(\mathbf{m}_i \mid (x_i, y_i), \theta)}{P(\mathbf{m}_i \mid (x_i, y_i), \theta)}\right] \tag{13}$$

Now, considering the outcome of Equation 3 , we can further derive it to

$$\mathrm{KL}(Q\|P) \tag{14}$$

$$= E_{Q_\phi}\left[\log \frac{Q_\phi(\mathbf{m}_i \mid (x_i, y_i), \theta)}{P_{\mathbf{m}_i \odot \theta}(y_i \mid x_i) \, P(\mathbf{m}_i \mid \theta)/P_\theta(y_i \mid x_i)}\right] \tag{15}$$

$$= -E_{Q_\phi}\left[\log P_{\mathbf{m}_i \odot \theta}(y_i \mid x_i)\right] + E_{Q_\phi}\left[P_\theta(y_i \mid x_i)\right] \tag{16}$$

$$+ \mathrm{KL}\left(Q_\phi(\mathbf{m}_i \mid (x_i, y_i), \theta) \, \| \, P(\mathbf{m}_i \mid \theta)\right) \tag{17}$$

Table 5: Mean Saliency Shift (MSS) between the original and unlearned models on the forget and retain sets. Results are reported as mean ± standard deviation across evaluation samples. Lower retain-set MSS indicates stronger preservation of the original attribution patterns.

| Method | MNIST | | FMNIST | | CIFAR10 | | ImageNet100 | |
|---|---|---|---|---|---|---|---|---|
| | Forget | Retain | Forget | Retain | Forget | Retain | Forget | Retain |
| Random Labels | $0.08 \pm 0.01$ | $0.04 \pm 0.02$ | $0.11 \pm 0.02$ | $0.09 \pm 0.03$ | $0.08 \pm 0.02$ | $0.09 \pm 0.02$ | $0.01 \pm 0.00$ | $0.02 \pm 0.01$ |
| Catastrophic Forgetting | $0.11 \pm 0.02$ | $0.05 \pm 0.01$ | $0.09 \pm 0.02$ | $0.09 \pm 0.02$ | $0.11 \pm 0.02$ | $0.03 \pm 0.01$ | $0.01 \pm 0.01$ | $0.02 \pm 0.01$ |
| Exact Unlearning | $0.15 \pm 0.01$ | $0.11 \pm 0.02$ | $0.13 \pm 0.02$ | $0.11 \pm 0.02$ | $0.11 \pm 0.02$ | $0.02 \pm 0.01$ | $0.01 \pm 0.01$ | $0.01 \pm 0.00$ |
| NegGrad+ | $0.15 \pm 0.01$ | $0.11 \pm 0.02$ | $0.10 \pm 0.02$ | $0.10 \pm 0.02$ | $0.06 \pm 0.03$ | $0.06 \pm 0.02$ | $0.02 \pm 0.01$ | $0.02 \pm 0.01$ |
| L1-Sparse | $0.13 \pm 0.01$ | $0.06 \pm 0.01$ | $0.12 \pm 0.02$ | $0.11 \pm 0.02$ | $0.11 \pm 0.02$ | $0.02 \pm 0.01$ | $0.01 \pm 0.00$ | $0.01 \pm 0.00$ |
| Saliency Unlearning | $0.14 \pm 0.02$ | $0.06 \pm 0.02$ | $0.12 \pm 0.02$ | $0.11 \pm 0.02$ | $0.12 \pm 0.02$ | $0.11 \pm 0.02$ | $0.04 \pm 0.01$ | $0.05 \pm 0.01$ |
| SSD | $0.08 \pm 0.01$ | $0.02 \pm 0.01$ | $0.11 \pm 0.02$ | $0.09 \pm 0.02$ | $0.09 \pm 0.02$ | $0.04 \pm 0.01$ | $0.00 \pm 0.00$ | $0.00 \pm 0.00$ |
| SCRUB | $0.16 \pm 0.01$ | $0.06 \pm 0.01$ | $0.10 \pm 0.02$ | $0.10 \pm 0.02$ | $0.10 \pm 0.02$ | $0.01 \pm 0.01$ | $0.01 \pm 0.00$ | $0.01 \pm 0.00$ |
| MIMU | $0.08 \pm 0.01$ | $0.04 \pm 0.01$ | $0.07 \pm 0.02$ | $0.05 \pm 0.02$ | $0.08 \pm 0.03$ | $0.01 \pm 0.01$ | $0.02 \pm 0.01$ | $0.01 \pm 0.00$ |

Note, the term $\log P_\theta(y_i \mid x_i)$ is constant as it doesn't rely on $m_i$. Finally, to enforce the sparsity constraint $\|m_i\|_0 \le \delta$, we incorporate a Lagrange multiplier term, leading to the complete objective:

$$\mathrm{argmin}_\phi \max_\lambda -E_{Q_\phi}[\log P_{\mathbf{m}_i \odot \theta}(y_i|x_i)]+$$
$$KL[Q_\phi(\mathbf{m}_i|(x_i, y_i), \theta)||P(\mathbf{m}_i|\theta)] + \lambda(||\mathbf{m}_i||_0 - \delta), \tag{18}$$

## B  Benchmark Datasets and Model Architecture

We evaluate our method on a diverse set of benchmark datasets that span different data modalities, input dimensionalities, and model complexities, as summarized in Table 4. Specifically, we include MNIST and Fashion-MNIST as canonical grayscale image classification benchmarks with relatively low-dimensional inputs, paired with lightweight MLP and CNN architectures to reflect common practice. CIFAR-10 serves as a standard natural image benchmark with higher visual complexity, for which we adopt a ResNet18 backbone following the original design, augmented with an additional hidden layer before the classification head as detailed in Table 4. To further assess scalability and behavior in large-scale visual settings, we include ImageNet-100, a widely used subset of ImageNet, and train a ResNet50 model to capture deeper hierarchical representations.

## C  Saliency-Map Analysis

To complement the qualitative visualizations, we quantify the change in input-attribution patterns before and after unlearning. For a sample $x$ with ground-truth label $y$, the gradient-based saliency map of a model parameterized by $\theta$ is defined as

$$S_\theta(x, y) = \mathcal{N}\left(\max_c \left|\frac{\partial z_y(x; \theta)}{\partial x_c}\right|\right), \tag{19}$$

where $z_y(x; \theta)$ is the logit associated with class $y$, $c$ indexes the input channels, and $\mathcal{N}(\cdot)$ denotes min–max normalization to $[0, 1]$. We use the same ground-truth target for the original and unlearned models so that the comparison captures changes in the attribution pattern for the same class, independently of changes in the predicted label. For each sample, we compute the *Mean Saliency Shift* (MSS) as

$$D_{\mathrm{sal}}(x) = \frac{1}{HW} \sum_{i=1}^{H} \sum_{j=1}^{W} |S_{\theta_\mathrm{u}}(x, y)_{ij} - S_{\theta_\mathrm{o}}(x, y)_{ij}|, \tag{20}$$

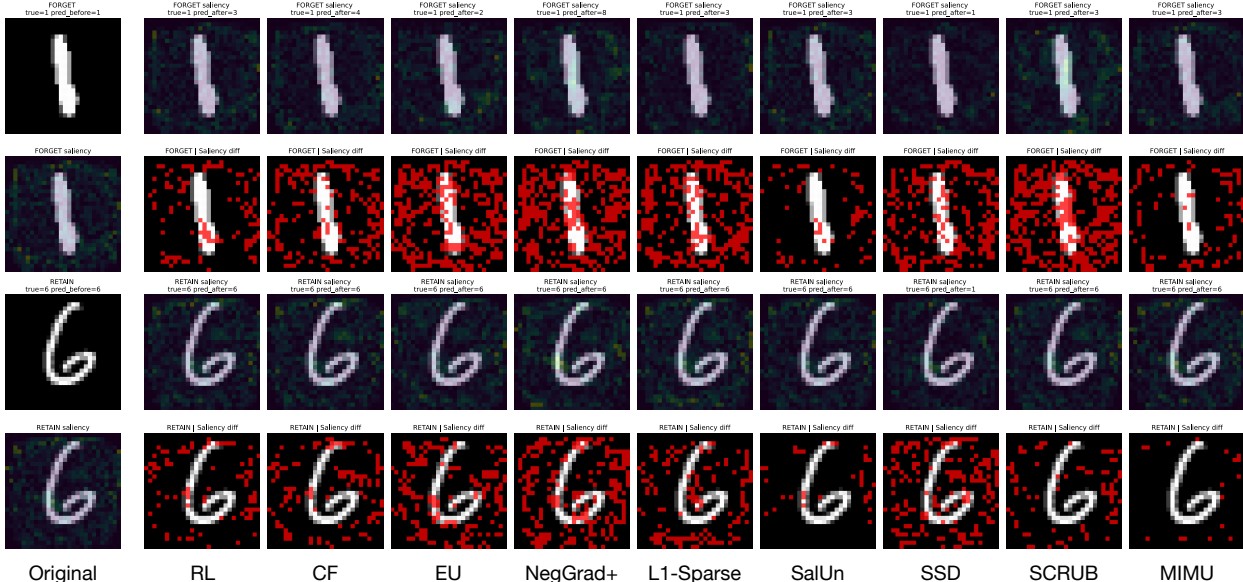

Figure 7: Extra saliency map analysis before and after unlearning on MNIST dataset. The top row shows the saliency map of a forget sample after unlearning. The second row shows the saliency difference between before and after unlearning for the forget sample. The bottom two rows show the corresponding results for a retain sample. Red dots highlight regions with notable saliency differences, which implies prediction pattern shift of target model on the data.

where $\theta_{\mathrm{o}}$ and $\theta_{\mathrm{u}}$ denote the parameters of the original and unlearned models, respectively, and $H$ and $W$ are the spatial dimensions of the saliency map. Lower values indicate stronger preservation of the original attribution pattern, whereas higher values indicate a greater redistribution of saliency after unlearning. We report the average MSS separately over the forget set $\mathcal{F}$ and retain set $\mathcal{R}$:

$$\bar{D}_{\mathrm{sal}}^{\mathcal{F}} = \frac{1}{|\mathcal{F}|} \sum_{x \in \mathcal{F}} D_{\mathrm{sal}}(x), \qquad \bar{D}_{\mathrm{sal}}^{\mathcal{R}} = \frac{1}{|\mathcal{R}|} \sum_{x \in \mathcal{R}} D_{\mathrm{sal}}(x). \tag{21}$$

The MSS results are presented in Table 5. MIMU consistently shows higher average MSS on the forget set with smaller MSS on the retain set.

## D  Hyperparameters

In Table 6, we provide detailed hyperparameter settings for the vision model, GCN training, baselines methods and MIMU masking. For baselines methods, the parameters reported in their original papers were used, and were tuned if not performing correctly. Additionally, we provide the details of GCN architecture in Table 7, and the complete training configuration and hyperparameters for each evaluated model in Table 8.

## E  Decomposition of Runtime During Inference

MIMU does not require inference-time optimization; however, processing an unlearning request involves several computational steps. The overall cost can be separated into a one-time offline training cost and a per-request unlearning cost.

The offline training cost, denoted by $T_{\mathrm{train}}$, includes constructing the parameter graph, computing node features, and performing repeated forward and backward passes to train the GCN-based control module. This cost is incurred only once, and the trained control module can subsequently be reused for any number of unlearning requests.

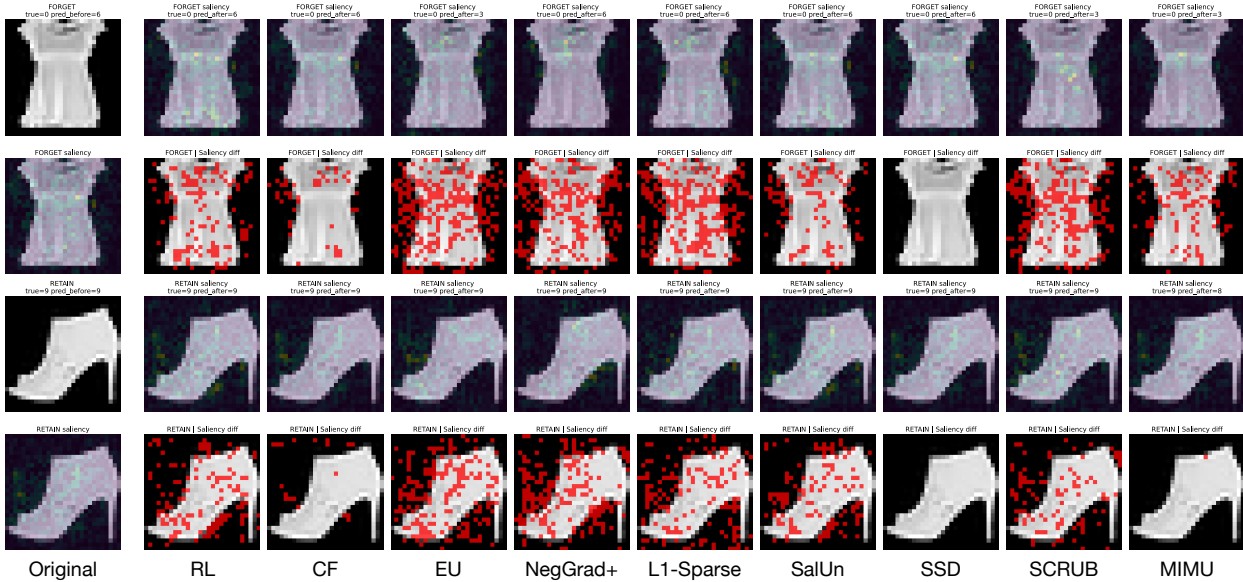

Figure 8: Extra saliency map analysis before and after unlearning on FMNIST dataset. The top row shows the saliency map of a forget sample after unlearning. The second row shows the saliency difference between before and after unlearning for the forget sample. The bottom two rows show the corresponding results for a retain sample. Red dots highlight regions with notable saliency differences, which implies prediction pattern shift of target model on the data.

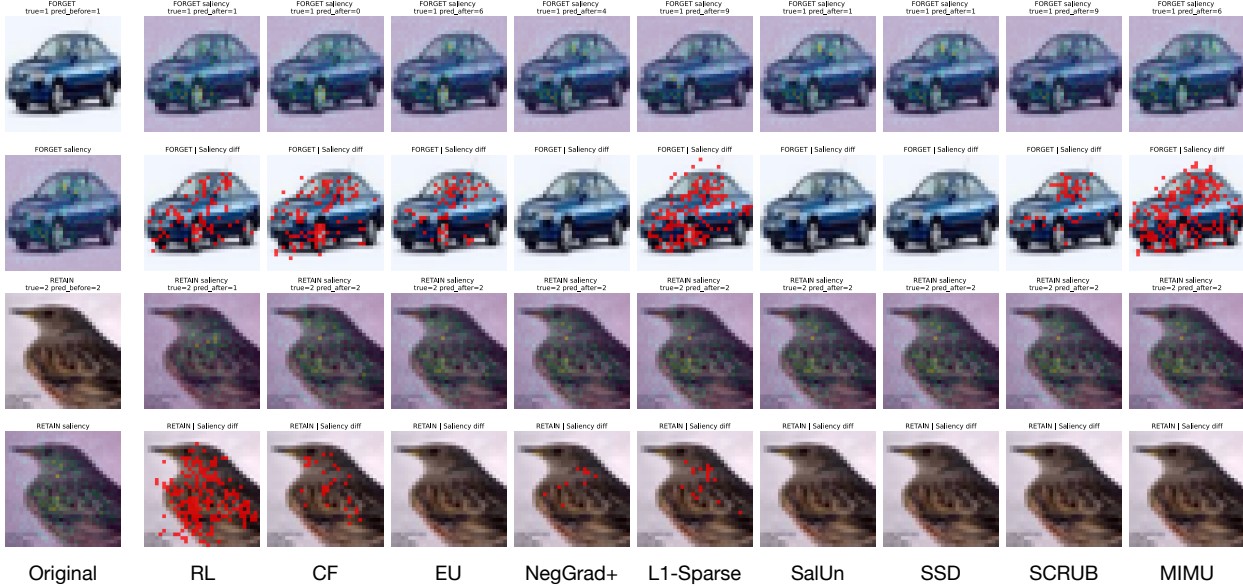

Figure 9: Extra saliency map analysis before and after unlearning on CIFAR10 dataset. The top row shows the saliency map of a forget sample after unlearning. The second row shows the saliency difference between before and after unlearning for the forget sample. The bottom two rows show the corresponding results for a retain sample. Red dots highlight regions with notable saliency differences, which implies prediction pattern shift of target model on the data.

For each request, the primary computational cost arises from feature extraction, mask generation, and mask application. Feature extraction requires one forward and one backward pass through the original vision model to obtain the gradients, weights, and activations used as input to the GCN. The GCN then generates a mask through a single forward pass, using a user-specified number of parameter indices, $k$ for parameter

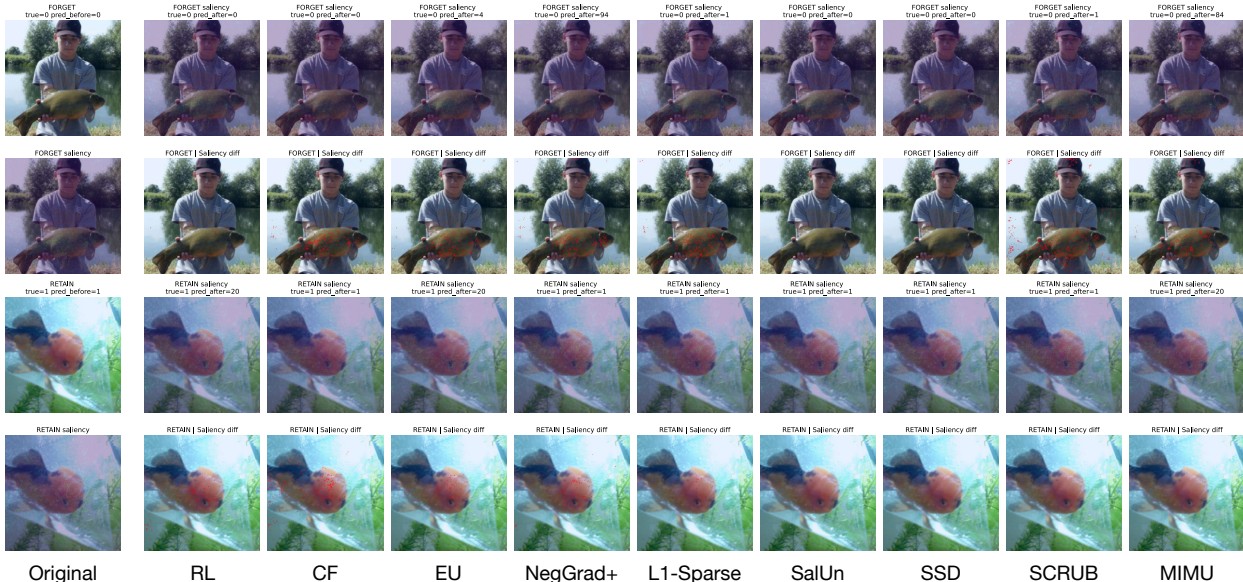

Figure 10: Extra saliency map analysis before and after unlearning on ImageNet100 dataset. The top row shows the saliency map of a forget sample after unlearning. The second row shows the saliency difference between before and after unlearning for the forget sample. The bottom two rows show the corresponding results for a retain sample. Red dots highlight regions with notable saliency differences, which implies prediction pattern shift of target model on the data.

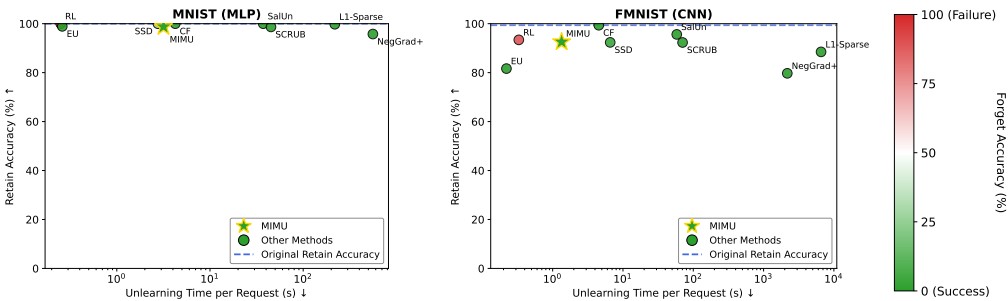

Figure 11: For every unlearning request, MIMU delivers the competitive results quickly. Other common MU techniques deliver similar performance but can cost significantly more in compute time. The graph above shows the Retain Accuracy in percentage on the Y axis, the Unlearning Time measured in wall-clock seconds on the X axis and the Forget Accuracy is measure from the hue of the marker, from red (failure - up to 100% retention) to green (success - down to 0% retention). Time values can be found in Table 9.

selection. Since $k$ is provided before the unlearning procedure, no additional search or optimization over $k$ is required.

A separate mask is generated for each representative sample from the forget set. Consequently, per sample representative, feature extraction and GCN inference are performed. The user specifies the number of forget-set representatives, denoted by $N_r^f$, before unlearning. MIMU does not require representative samples from the retain set.

Finally, the generated masks for all forget-set representatives are combined using their union and then applied to the selected model layer. This step consists of an element-wise multiplication between the aggregated mask and the parameters of that layer.

For (R) unlearning requests, the amortized cost per request is

Table 6: Comprehensive Hyperparameter Configurations for Unlearning Experiments. A hyphen means the default parameter was used for class regime, or the corresponding class parameter was used for subclass regime.

| Category | Hyperparameter | Default | MNIST (MLP) | | FMNIST (CNN) | | CIFAR10 (ResNet18) | | ImageNet100 (ResNet50) | |
|---|---|---|---|---|---|---|---|---|---|---|
| | | | Class | Subclass | Class | Subclass | Class | Subclass | Class | Subclass |
| **Vision Training** | Epochs | 100 | - | - | 40 | - | 200 | - | 20 | - |
| | Learning Rate | $3 \times 10^{-4}$ | - | - | $10^{-3}$ | - | 0.1 | - | 0.01 | - |
| | Batch Size | 256 | - | - | - | - | - | - | 128 | - |
| | Weight Decay | 0.01 | - | - | - | - | $5 \times 10^{-4}$ | - | $10^{-4}$ | - |
| | Optimizer | AdamW | - | - | - | - | SGD | - | SGD | - |
| | Opt. Momentum | 0.9 | - | - | - | - | - | - | - | - |
| | LR Scheduler | None | - | - | - | - | Cosine | - | Cosine | - |
| **GCN Training** | Epochs | 11 | - | - | 12 | - | 8 | - | 8 | - |
| | Learning Rate | $10^{-3}$ | - | - | - | - | - | - | - | - |
| | mask_K (Utility) | 0.1 | - | - | 0.05 | - | - | - | 0.01 | - |
| | Sample Rate | 1.0 | 0.05 | - | 0.05 | - | 0.05 | - | 0.01 | - |
| | Hidden Dim | 32 | - | - | - | - | - | - | - | - |
| | Gumbel Temp | 0.1 | - | - | - | - | - | - | - | - |
| **CF** | Steps | 50 | 2000 | - | 2000 | - | 500 | - | 10000 | - |
| | Learning Rate | $10^{-3}$ | 0.02 | - | 0.02 | - | 0.02 | - | 0.002 | - |
| **EU** | Steps | 50 | 1000 | - | 1000 | - | - | - | 10000 | - |
| | Learning Rate | $10^{-3}$ | - | - | - | - | - | - | - | - |
| **RL** | Steps | 50 | 1000 | - | 1000 | - | - | - | 1000 | - |
| | Learning Rate | $10^{-3}$ | - | - | - | - | - | - | - | - |
| **NegGrad+** | Alpha ($\alpha$) | 0.9 | 0.99 | - | 0.99 | - | 0.99 | - | 0.99 | - |
| | Steps | 10 | 200 | - | 200 | - | 20 | - | - | - |
| | Weight Decay | 0.1 | - | - | - | - | - | - | - | - |
| **L1-Sparse** | Alpha ($\alpha$) | 0.2 | 0.01 | - | 0.005 | - | 0.005 | - | 0.005 | - |
| | Steps | 50 | 100 | - | 100 | - | - | - | - | - |
| | Warmup | 5 | - | - | - | - | - | - | - | - |
| **SalUn** | Threshold | 0.15 | - | - | 0.2 | - | 0.5 | - | - | - |
| | Momentum | 1.0 | 0.9 | - | 0.9 | - | - | - | 0.9 | - |
| **SSD** | Dampening ($\lambda$) | 1.0 | - | - | - | - | - | - | - | - |
| | Selection ($\alpha$) | 10.0 | 2.0 | 2.0 | 20.0 | 20.0 | 5.0 | 5.0 | - | - |
| **SCRUB** | SGDA Epochs | 10 | - | - | - | - | - | - | - | - |
| | Learning Rate | $5 \times 10^{-4}$ | - | - | - | - | - | - | - | - |
| | M-Steps | 3 | - | - | - | - | - | - | - | - |
| | Gamma ($\gamma$) | 1.0 | - | - | - | - | - | - | - | - |
| | Alpha ($\alpha$) | 0.5 | - | - | - | - | - | - | - | - |
| | Beta ($\beta$) | 0.0 | - | - | - | - | - | - | - | - |
| | KD Temp ($T$) | 2 | - | - | - | - | - | - | - | - |
| **MIMU Eval** | topK (Pruning) | 0.1 | 0.05 | - | 0.0025 | 0.002 | 0.03 | 0.002 | $10^{-3}$ | $2 \times 10^{-4}$ |
| | Representatives | 100 | 300 | - | 600 | - | - | - | - | - |
| | Batch Size | 32 | - | - | - | - | - | - | - | - |

Table 7: Comprehensive detail of GCN model architecture configurations. $V = M \cdot N$ is the number of nodes (weight parameters in the mask layer). These specifications provide the implementation details required to reproduce the base GCN controller.

| Component | Type | In dim | Out dim | Bias | Activation |
|---|---|---|---|---|---|
| Input projection | GCNConv | 4 | $d$ | No | — |
| Hidden layer 1 | GCNConv | $d$ | $d$ | No | Sigmoid |
| Hidden layer 2 | GCNConv | $d$ | $d$ | No | Sigmoid |
| Output projection | nn.Linear | $d$ | 1 | Yes | — (logits) |
| Squeeze | .squeeze() | $[V, 1]$ | $[V]$ | — | — |

$$T_{\text{amortized}} = \frac{T_{\text{train}} + R T_{\text{req}}}{R} \quad = \frac{T_{\text{train}}}{R} + T_{\text{req}}, \tag{22}$$

Table 8: GCN training hyperparameters across experimental settings. R18 = ResNet-18; R50 = ResNet-50; IN100 = ImageNet-100. The details fully specify the GCN training configuration used in across experiments for each model.

| Hyperparameter | Default | MNIST-MLP | FMNIST-CNN | CIFAR10-R18 | IN100-R50 |
|---|---|---|---|---|---|
| GCN epochs | 11 | 11 | 12 | 8 | 8 |
| Learning rate | 1e-3 | 1e-3 | 1e-3 | 1e-3 | 1e-3 |
| Weight decay | 5e-4 | 5e-4 | 5e-4 | 5e-4 | 5e-4 |
| Optimizer | AdamW | AdamW | AdamW | AdamW | AdamW |
| Batch size | 64 | 64 | 64 | 64 | 64 |
| GCN training dataset size | 2048 | 2048 | 2048 | 2048 | 2048 |
| Hidden dimension $d$ | 32 | 32 | 32 | 32 | 32 |
| Number of layers | 3 | 3 | 3 | 3 | 3 |
| Gumbel temperature $\tau$ | 0.1 | 0.1 | 0.1 | 0.1 | 0.1 |
| Train-time mask $K^1$ | 10% | 2% | 5% | 10% | 1% |
| Graph sample rate | 1.0 | 1.0 | 0.05 | 0.05 | 0.01 |
| Eval prune $K$ | 10% | 2% | 0.25% | 10% | 0.1% |

Table 9: Unlearning time per request (wall-clock, in seconds). Lower values represent faster unlearning requests. Results are averaged over 5 independent runs. MIMU clearly demonstrates rapid unlearning across architectures.

| Dataset | RL | CF | EU | NegGrad+ | L1-Sparse | SalUn | SSD | SCRUB | MIMU |
|---|---|---|---|---|---|---|---|---|---|
| MNIST (MLP) | $0.25 \pm 0.01$ | $4.24 \pm 0.39$ | $0.26 \pm 0.01$ | $559.20 \pm 2.68$ | $218.40 \pm 3.29$ | $37.24 \pm 3.98$ | $2.76 \pm 0.10$ | $45.12 \pm 0.26$ | $3.17 \pm 0.21$ |
| FMNIST (CNN) | $0.33 \pm 0.15$ | $4.53 \pm 0.07$ | $0.22 \pm 0.01$ | $2174.17 \pm 53.61$ | $6572.42 \pm 58.42$ | $58.24 \pm 0.91$ | $6.58 \pm 0.00$ | $70.43 \pm 0.90$ | $1.34 \pm 0.20$ |
| CIFAR10 (ResNet 18) | $1.69 \pm 0.77$ | $5.88 \pm 1.58$ | $1.34 \pm 0.25$ | $282.43 \pm 106.47$ | $406.42 \pm 102.06$ | $140.75 \pm 0.24$ | $67.62 \pm 4.65$ | $220.54 \pm 79.43$ | $2.25 \pm 0.25$ |
| ImageNet100 (ResNet 50) | $2.47 \pm 0.53$ | $141.82 \pm 24.61$ | $148.93 \pm 31.92$ | $2035.78 \pm 722.72$ | $7017.55 \pm 1167.38$ | $1948.37 \pm 0.78$ | $356.86 \pm 31.98$ | $3394.54 \pm 908.64$ | $9.18 \pm 2.08$ |

where $T_{\text{req}}$ denotes the computational cost of a single unlearning request. It is determined by the feature-extraction cost $T_{\text{feat}}$, the cost of a GCN forward pass $T_{\text{GCN}}$, the mask-application $T_{\text{apply}}$, and the number of forget-set representatives $N_r^f$:

$$T_{\text{req}} = N_r^f \left( T_{\text{feat}} + T_{\text{GCN}} \right) + T_{\text{apply}}. \tag{23}$$

## F   Gumbel Top-K Sampling for MIMU

Our implementation of Gumbel Top-K sampling follows the subset sampling paradigm of Xie & Ermon (2021) with the Gumbel Straight-Through Estimator of Jang et al. (2017). This implementation generates an unranked set of parameters to be masked during unlearning. During training the forward pass uses hard (binary) top-k masks, while the backward pass employs the Gumbel Straight-Through estimator. The Gumbel Straight-Through Estimator is used to preserve the gradients and allow for training with hard masks. The Gumbel temperature $\tau$ is held constant throughout training. No temperature annealing is performed. At inference time, a final hard mask is generated for a given sample by the GCN from a feedforward process through the model. Further information about the inference time process is given in Appendix E. Implementation code in PyTorch is included below:

```
1    top_k = math.ceil(len(logits) * k)
2    gumbel_noise = -torch.log(-torch.log(torch.rand_like(logits) + eps) + eps)
3    gumbel_logits = (logits + gumbel_noise) / temperature
4    soft_mask = F.softmax(gumbel_logits, dim=-1)
5    topK_indices = gumbel_logits.topk(k=top_k).indices
6    hard_mask = torch.zeros_like(gumbel_logits, dtype=torch.float)
7    hard_mask[topK_indices] = 1.0
8    final_mask = (hard_mask - soft_mask).detach() + soft_mask
```

Table 10: Ablation study of the inputs to the GCN. Results are averaged over 5 runs with standard deviation error bars.

| GCN Input Ablation | MNIST | | FMNIST | | CIFAR10 | | ImageNet100 | |
|---|---|---|---|---|---|---|---|---|
| | Forget | Retain | Forget | Retain | Forget | Retain | Forget | Retain |
| Original GCN | $0.78 \pm 1.32$ | $98.78 \pm 0.36$ | $0.73 \pm 0.52$ | $92.55 \pm 0.46$ | $0.00 \pm 0.00$ | $99.64 \pm 0.02$ | $0.00 \pm 0.00$ | $90.72 \pm 0.08$ |
| No In Activations | $1.72 \pm 2.47$ | $96.55 \pm 0.28$ | $20.31 \pm 3.45$ | $92.91 \pm 0.57$ | $3.42 \pm 1.11$ | $99.73 \pm 0.02$ | $0.00 \pm 0.00$ | $89.88 \pm 0.08$ |
| No Out Activations | $0.00 \pm 0.00$ | $90.46 \pm 1.81$ | $0.01 \pm 0.01$ | $84.93 \pm 2.17$ | $0.00 \pm 0.00$ | $99.72 \pm 0.02$ | $0.00 \pm 0.00$ | $89.94 \pm 0.19$ |
| No Weights | $0.04 \pm 0.03$ | $95.71 \pm 0.35$ | $42.47 \pm 8.76$ | $92.65 \pm 0.65$ | $0.02 \pm 0.06$ | $99.71 \pm 0.01$ | $57.40 \pm 2.52$ | $91.47 \pm 0.02$ |
| No Gradients | $0.05 \pm 0.06$ | $93.44 \pm 1.44$ | $71.72 \pm 6.86$ | $86.85 \pm 1.90$ | $3.63 \pm 1.27$ | $99.65 \pm 0.02$ | $19.26 \pm 2.71$ | $88.89 \pm 0.06$ |

Mask $\kappa$ sets the number of model parameters selected for masking. $\kappa$ and $\tau$ are both percentages given by the user as a hyperparameter at the outset of the unlearning process. The particular settings we used in our experiments is given in Table 6.

## G   Additional Experimental Plots and Tables

Figures 7, 8, 9 and 10 present additional saliency map analysis results on the MNIST,CIFAR-10 and ImageNet-100 datasets. Consistent with the main paper, most existing approximate MU methods induce significant prediction pattern changes after unlearning, whereas MIMU and SalUn generally preserve the model's original prediction behavior and cause less disruption. Such the significancy of such observation varies based on the dataset: On MNIST and FMNIST datasets, the corruption of prediction pattern is more clear than that of the CIFAR-10 and ImageNet datasets.

Figure 11 provides additional analysis of the unlearning effectiveness and computational efficiency on MNIST and FMNIST, complementing the results presented in the main paper. The x-axis represents the average wall-clock unlearning time required to process a single request, while the y-axis reports the retain accuracy after unlearning. The color gradient indicates the forget accuracy, where green shades correspond to lower forget accuracy and therefore more effective forgetting. The dashed horizontal line denotes the accuracy of the original model prior to unlearning. MIMU is highlighted using star markers, whereas competing methods are represented by circular markers.

Table 9 reports the average wall-clock execution time (in seconds) required by each machine unlearning method, measured over five independent runs. The experiments tested MIMU and other unlearning methods on the MLP (MNIST), CNN (FMNIST), ResNet-18 (CIFAR-10), and ResNet-50 (ImageNet-100).

## H   Ablation Studies

**MIMU Hyperparameters** We conduct an ablation study to analyze the impact of two key hyperparameters in MIMU: (1) the number of representative samples and (2) the top-$\kappa$ masking threshold determined by the top-$\kappa$ weights selected for unlearning.

Figure 12 presents the results on MNIST. We observe that MIMU achieves effective unlearning even when using a relatively small number of representative samples and a small top-$\kappa$ threshold. Simultaneously, the method drives the forget-set accuracy close to zero while preserving the retain-set accuracy. MNIST sees the most deterioration of the retain accuracy with increased top-$\kappa$ thresholds.

The results for FMNIST are shown in Figure 13. Similar to CIFAR-10, MIMU causes a substantial reduction in forget-set accuracy while maintaining the retain-set performance. Similar to ImageNet-100, FMNIST requires very small top-$\kappa$ thresholds to achieve competitive unlearning performance.

**GCN Inputs**: We conducted an ablation study on the inputs to GCN training to determine how the different inputs affect the GCN's abilities to find suitable unlearning masks. Table 10 shows the effects that removing each of the inputs has on the final output.

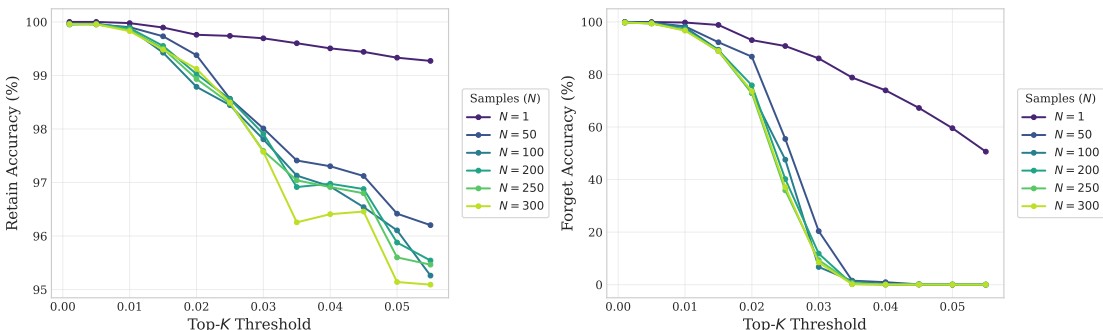

Figure 12: Ablation study of the number of representative samples and top-$\kappa$ masking threshold in MIMU. Results show class-level unlearning performance on MNIST, measured in terms of retain-set and forget-set accuracies after unlearning.

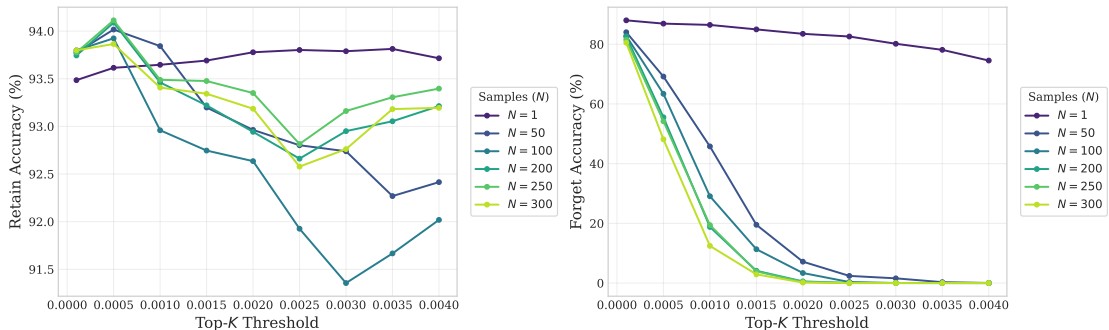

Figure 13: Ablation study of the number of representative samples and top-$\kappa$ masking threshold in MIMU. Results show class-level unlearning performance on FMNIST, measured in terms of retain-set and forget-set accuracies after unlearning.

# I    Discussion: Scalability of MIMU

As stated in the main paper (Section 3.2), using a graph-convolution-based model as the control function can be challenging when scaling to models with millions or billions of parameters. Fortunately, prior work (Fel et al., 2024; Dorszewski et al., 2025) has shown that early layers of neural networks primarily act as generic feature extractors and contribute less directly to specific predictions. This insight allows us to reduce computational overhead by focusing the masking and unlearning process on the last one or two layers of the network, where the parameters have the most direct influence on predictions while involving far fewer parameters. In our experiments, we show that MIMU can effectively unlearn data from a ResNet-50 model (25.6 million parameters) on ImageNet-100, demonstrating the scalability of the proposed method in terms of empirical effectiveness.

The same logic is transferable to transformer models, where recent research suggests that a model's predictive ability is strongly tied to the linear layers between attention blocks. Accordingly, MIMU can control or mask neurons in these linear layers to suppress the model's predictive ability on the forget set.

As an additional note, the aforementioned scalability limitation applies only to the GCN-based control unit described in the main paper. In this paper, we propose MIMU as a framework that learns a control unit capable of performing unlearning without request-time optimization. The control unit can be any trainable model. A simple neural network-based control unit can also serve as the control unit, although its performance may be sacrificed, as demonstrated in Section J. Replacing the GCN with another, more scalable control model would allow the MIMU framework to be further scaled up.

Table 11: Control module training time (wall-clock, in seconds). Lower values represent faster training time. Results are averaged over 5 independent runs. Results are averaged over 5 runs with standard deviation error bars.

| Method | MNIST | FMNIST | CIFAR10 | ImageNet100 |
|---|---|---|---|---|
| MIMU (GCN) | $25.52 \pm 0.56$ | $30.94 \pm 0.94$ | $170.52 \pm 0.29$ | $868.59 \pm 1.91$ |
| MIMU (MLP) | $14.86 \pm 0.26$ | $19.83 \pm 0.11$ | $40.65 \pm 0.27$ | $181.76 \pm 1.78$ |

Table 12: Class-level unlearning performance with MLP control module in terms of post-unlearning prediction accuracy. Results are averaged over 5 runs with standard deviation error bars.

| Method | MNIST | | FMNIST | | CIFAR10 | | ImageNet100 | |
|---|---|---|---|---|---|---|---|---|
| | Forget | Retain | Forget | Retain | Forget | Retain | Forget | Retain |
| MIMU (GCN) | $0.78 \pm 1.32$ | $98.78 \pm 0.36$ | $0.73 \pm 0.52$ | $92.55 \pm 0.46$ | $0.00 \pm 0.00$ | $99.64 \pm 0.02$ | $0.00 \pm 0.00$ | $90.72 \pm 0.08$ |
| MIMU (MLP) | $0.00 \pm 0.00$ | $94.19 \pm 0.79$ | $0.01 \pm 0.01$ | $89.93 \pm 1.38$ | $0.00 \pm 0.00$ | $99.25 \pm 0.14$ | $0.00 \pm 0.00$ | $90.21 \pm 0.12$ |

## J    Control Module Alternative Architectures

A graph-convolution-based model is not the only possible architecture for a suitable control module in MIMU. Tables 11 and 12 demonstrate our ablation study of a MLP control module in place of a GCN. As shown, the MLP offers requires less training time and also shows competitive unlearning efficacy on the forget set but does not maintain the level of retain set performance that the GCN achieves.

## K    Membership Inference Attack

Table 13 shows the results of a subclass-level membership inference attack (MIA) Shokri et al. (2017) on the unlearned models. The Threshold method is a threshold attack that uses the entropy of the model outputs to determine which samples are included in the training set. The Classifier method is a logistic regression classifier trained on the outputs, loss, and entropy of the model.

The results show that MIAs achieve poor attack performance against almost all evaluated unlearning methods, which, when considered in isolation, could be interpreted as evidence of successful unlearning. However, Table 2 shows that many of these methods substantially degrade performance on the retain set while reducing accuracy on the forget set. Thus, a low MIA attack success rate does not necessarily indicate that a method has selectively removed information associated with the forget set while preserving the model's remaining utility. Instead, it may partly result from broader deterioration of the model's predictive behavior. That said, these results should be read alongside Table 2 to understand which MIA successes are the result of increased damage to the unlearned model's performance.

## L    Limitations

This work focuses on a control-based machine unlearning (MU) framework that achieves minimally invasive unlearning via learned masking, without inference-time optimization or weight updates. In the current implementation, the controller module is instantiated as a graph convolutional network (GCN), which efficiently captures parameter interactions and remains practical as model size scales. However, GCNs are known to be sensitive to training configurations Cong et al. (2021), particularly architectural choices such as network depth and the number of message-passing steps, which can pose practical challenges for less experienced practitioners. Importantly, this limitation is not intrinsic to the proposed MIMU framework. The controller is modular and can be replaced with alternative neural architectures beyond GCNs. Investigating other controller designs (e.g., transformer- or diffusion-based models) to reduce manual calibration while preserving optimization-free unlearning is a promising direction for future work.

Table 13: Subclass-level unlearning performance in terms of Membership Inference Attack Receiver Operating Characteristic Area Under the Curve (ROC AUC). Attacks where conducted with an entropy threshold and a logistic classifier. Results are averaged over 5 runs with standard deviation error bars.

| Method | MNIST | | FMNIST | | CIFAR10 | | ImageNet100 | |
|---|---|---|---|---|---|---|---|---|
| | Threshold | Classifier | Threshold | Classifier | Threshold | Classifier | Threshold | Classifier |
| Random Labels | $0.37 \pm 0.00$ | $0.50 \pm 0.00$ | $0.58 \pm 0.00$ | $0.59 \pm 0.01$ | $0.44 \pm 0.00$ | $0.50 \pm 0.00$ | $0.62 \pm 0.00$ | $0.68 \pm 0.03$ |
| Catastrophic Forgetting | $0.50 \pm 0.00$ | $0.50 \pm 0.00$ | $0.54 \pm 0.00$ | $0.54 \pm 0.01$ | $0.50 \pm 0.00$ | $0.50 \pm 0.00$ | $0.46 \pm 0.00$ | $0.59 \pm 0.02$ |
| Exact Unlearning | $0.55 \pm 0.00$ | $0.55 \pm 0.00$ | $0.57 \pm 0.00$ | $0.61 \pm 0.01$ | $0.49 \pm 0.00$ | $0.50 \pm 0.00$ | $0.39 \pm 0.00$ | $0.48 \pm 0.04$ |
| NegGrad+ | $0.51 \pm 0.00$ | $0.21 \pm 0.00$ | $0.53 \pm 0.00$ | $0.52 \pm 0.00$ | $0.52 \pm 0.00$ | $0.53 \pm 0.00$ | $0.56 \pm 0.00$ | $0.37 \pm 0.02$ |
| L1-Sparse | $0.37 \pm 0.00$ | $0.37 \pm 0.00$ | $0.56 \pm 0.00$ | $0.55 \pm 0.00$ | $0.48 \pm 0.00$ | $0.51 \pm 0.00$ | $0.50 \pm 0.00$ | $0.50 \pm 0.02$ |
| Saliency Unlearning | $0.13 \pm 0.00$ | $0.14 \pm 0.00$ | $0.52 \pm 0.00$ | $0.51 \pm 0.00$ | $0.40 \pm 0.00$ | $0.41 \pm 0.00$ | $0.50 \pm 0.00$ | $0.56 \pm 0.04$ |
| SSD | $0.51 \pm 0.00$ | $0.52 \pm 0.00$ | $0.53 \pm 0.00$ | $0.54 \pm 0.01$ | $0.51 \pm 0.00$ | $0.51 \pm 0.00$ | $0.41 \pm 0.00$ | $0.50 \pm 0.03$ |
| SCRUB | $0.45 \pm 0.00$ | $0.45 \pm 0.00$ | $0.58 \pm 0.00$ | $0.57 \pm 0.00$ | $0.50 \pm 0.00$ | $0.50 \pm 0.00$ | $0.49 \pm 0.00$ | $0.52 \pm 0.02$ |
| MIMU | $0.13 \pm 0.00$ | $0.14 \pm 0.00$ | $0.51 \pm 0.04$ | $0.44 \pm 0.13$ | $0.48 \pm 0.02$ | $0.53 \pm 0.02$ | $0.65 \pm 0.09$ | $0.69 \pm 0.09$ |

## M    Broader Vision: Interpretability Through MIMU

Our framework has broader impact beyond efficient and sustainable machine unlearning, as the MIMU masking mechanism also offers a principled route toward explaining model predictions. By identifying a minimal subset of parameters whose masking reliably lowers model's predictive performance on the forget set MIMU implicitly characterizes which internal components are most responsible for specific behaviours, in line with masking-based explanation methods that attribute predictions to a sparse subset of parameters, inputs, or circuits. This dual use of masks for both unlearning and explanation can strengthen the evidence that minimally invasive interventions on a small, well-identified subnetwork are sufficient to control model behaviour, rather than requiring opaque, global parameter changes. In practice, this connection can support more transparent auditing pipelines in which the same machinery used to honour data-deletion requests (e.g., for regulatory compliance or user rights) also yields interpretable summaries of how and where sensitive information was encoded. More broadly, tying unlearning to sparse, explanation-driven interventions encourages the design of models whose internal structure is explicitly aligned with controllability and accountability, helping to bridge the gap between privacy-preserving ML and explainable AI.

