# OpenReview forum: "Minimally Invasive Machine Unlearning via Posterior Control"
_TMLR — Under review for TMLR_

### Review · Reviewer_cfea · 2026-06-20

**Summary Of Contributions:**

The paper proposes Minimally Invasive Machine Unlearning (MIMU), a method for efficient approximate unlearning in trained classification models. The main idea is to train a controller that predicts a sparse parameter mask for a given forget request. At unlearning time, the method masks a small number of parameters associated with the forget samples, aiming to reduce predictive accuracy/confidence on the forget set while preserving behavior on retained data. The paper formulates this as a posterior-control problem over masks, implements the controller with graph neural networks over selected model weights, and evaluates on MNIST, Fashion-MNIST, CIFAR-10, and ImageNet100 under class-level and subclass/random unlearning settings.

Key strengths are the clear motivation, the amortized request-time design, strong class-level unlearning results, and substantial runtime advantages over optimization-based baselines. Key weaknesses are that the evidence mainly supports predictive suppression rather than strict removal of training-data influence, subclass/random unlearning is much less convincing, and the method’s robustness under instance-level deletion, repeated requests, recovery attacks, and stronger privacy-oriented evaluations remains unclear.

**Audience:**

Yes

**Audience Explanation:**

Machine unlearning is an important topic, and the idea of amortizing parameter-mask selection into a learned controller is interesting. Researchers working on unlearning, model editing, model repair, sparse interventions, and deployment-time model governance would likely find the results useful, especially the evidence that sparse late-layer masking can be effective and fast for class-level unlearning.

The paper is also useful as an empirical study of the trade-off between forget-set suppression and retain-set preservation. Even if the current method does not fully solve fine-grained or privacy-oriented unlearning, the controller-based approach is a relevant direction.

**Claims And Evidence:**

No

**Claims Explanation:**

The evidence supports a narrower claim than the paper sometimes appears to make. The experiments are convincing that MIMU can often reduce forget-set accuracy/confidence at low request-time cost, especially for class-level unlearning in image classification. The retain accuracy and runtime results are useful, and the entropy/saliency analyses provide some additional support for the “minimally invasive” narrative.

However, the evidence is not sufficient for stronger claims about removing training-data influence or providing broadly reliable machine unlearning. The paper mainly evaluates forget/retain accuracy, entropy, and saliency maps. These do not establish equivalence to retraining, resistance to membership inference, resistance to relearning/recovery, or removal of instance-specific information. The subclass/random unlearning results are also much weaker, with high forget accuracy remaining in some settings. This suggests the method may work best when the forget target aligns with a relatively localized class-level decision structure, but is less reliable for fine-grained deletion.

The controller training cost, memory overhead, amortization break-even point, and behavior under multiple sequential unlearning requests are also not sufficiently characterized. These are important for a method whose main claim is low request-time cost and minimally invasive deployment.

**Requested Changes:**

## Critical

1. Narrow the claims. The paper should clearly distinguish “reducing predictive accuracy/confidence on the forget set” from “removing training-data influence” or “matching retraining.” The current evidence supports the former much more strongly than the latter.

2. Add stronger fine-grained unlearning evaluations. Instance-level or small-set unlearning should be included, ideally with membership inference or another privacy-oriented metric. The subclass/random results already show that fine-grained deletion is difficult, so this needs more direct analysis.

3. Add recovery/relearning tests. After applying MIMU, lightly fine-tune on retain data or unlabeled data and measure whether forget accuracy or confidence quickly recovers. This would help determine whether MIMU truly removes influence or only temporarily suppresses predictions.

---

> ### Author Response · Authors · 2026-07-14
>
> Thank you for reviewing this paper. We are glad that you feel the control-based unlearning approach we proposed in this paper is interesting and well-motivated. Below we address the questions and concerns you raised in your comments.
>
> **Narrow the claims and focus on "reducing predictive accuracy/confidence on the forget set" rather than "removing training data influence".**
>
> Thank you for the suggestion. In our paper, we emphasized that the focus of this work is to reduce predictive accuracy and confidence on the forget set while preserving model performance on the retain set, as stated in the second paragraph of Section 2.1 on page 3. Our experimental design also reiterates this claim in the first paragraph of Section 4.1 on page 3. The experimental results further reflect this intention, as acknowledged in your previous comments.
>
> We do not intend to overclaim that our method removes training data influence or matches full retraining. However, we understand your suggestion that this point should be made clearer. Therefore, we have revised the paper to remove ambiguous statements wherever applicable, and we have highlighted the changes in red. We are now confident that the paper does not claim to match retraining or to remove training data influence.
>
> **Instance-level or small-set unlearning should be included, ideally with membership inference or another privacy-oriented metric**.
>
> Thank you for your suggestion. We would like to highlight that instance-level or small-set unlearning remains a challenging problem in machine unlearning literature. Several related analytical works [1,2,3] suggest that this problem can be extremely difficult, partly due to the amortization effect of neural networks and the superposition of information across neurons. While we are also interested in addressing this important challenge, instance-level unlearning is not the focus or claim of the present paper. Instead, our work focuses on reducing the predictive confidence of the model on the forget set.
>
> Regarding the use of membership inference as an evaluation metric, we did consider it. However, membership inference is often treated as a proxy for the model’s predictive confidence, as discussed in [4,5,6]. And recent literature has criticized membership-inference-based evaluation due to its limited precision in reflecting the actual effectiveness of unlearning on a model [7,8,9]. For this reason, we instead directly report the entropy of the model’s predictions after unlearning in our paper, which more directly captures changes in predictive confidence.
>
> We understand the reviewer’s concern regarding the completeness of our experimental evaluation, as membership inference attacks (MIAs) have been used in several prior machine-unlearning studies despite their known limitations. To address this concern, we have added MIA-based experiments (on sub/random-class unlearning task) and a corresponding discussion to the revised manuscript. Please refer to Table 13 on page 10 of the appendix for details; for convenience, we also provide the main results below.
>
> The results show that MIAs achieve poor attack performance against almost all evaluated unlearning methods, which, when considered in isolation, could be interpreted as evidence of successful unlearning. However, Table 2 shows that many of these methods substantially degrade performance on the retain set while reducing accuracy on the forget set. Thus, a low MIA attack success rate does not necessarily indicate that a method has selectively removed information associated with the forget set while preserving the model’s remaining utility. Instead, it may partly result from broader deterioration of the model’s predictive behavior.
>
> Therefore, although we include MIA results to ensure a more complete comparison with prior work, we agree with recent literature that MIA-based evaluation can provide a biased or incomplete measure of unlearning quality and should not be interpreted independently of retain-set utility and other unlearning metrics.

---

> ### Author Response · Authors · 2026-07-14
>
> **About resistant to fine-tune-based predictive ability recovery**
>
> Our method focuses on minimally invasive modification of the original model, with the goal of achieving unlearning at low computational cost. Specifically, we aim to reduce the model’s predictive ability on the forget set by masking only a very small number of parameters through a control unit. This is substantially different from merely “temporarily suppressing prediction.”
>
> The recovery of predictive ability through fine-tuning does not imply that an unlearning method, including ours, only temporarily suppresses prediction. Rather, it reflects the parameter-level distance between two models [10] (before and after fine-tuning). Since our method affects only a small number of parameters, fine-tuning can relearn the relevant predictive ability within one or two epochs as expected. As a point of comparison, two epochs of training are often sufficient for a previously untrained model to acquire non-trivial predictive ability on typical data points. We are aware of a line of research on tamper-resistant unlearning [11], which specifically aims to prevent models from regaining predictive ability after unlearning. However, this represents a different research objective/interest from the one considered in our work.
>
>
>
> [1] Thudi, Anvith, et al. "On the necessity of auditable algorithmic definitions for machine unlearning." 31st USENIX security symposium (USENIX Security 22). 2022.
>
>  [2] Zhao, Kairan, et al. "What makes unlearning hard and what to do about it." Advances in Neural Information Processing Systems 37 (2024): 12293-12333
>
>  [3] Ginart, Antonio, et al. "Making ai forget you: Data deletion in machine learning." Advances in neural information processing systems 32 (2019)
>
> [4] Kodge, Sangamesh et al. “Deep Unlearning: Fast and Efficient Gradient-free Approach to Class Forgetting.” Transactions on Machine Learning Research (TMLR), 2024
>
> [5] Jia, Jinghan et al. “Model Sparsity Can Simplify Machine Unlearning.” Advances in Neural Information Processing Systems 36 (2023)
>
> [6] Golatkar, Aditya et al. “Forgetting Outside the Box: Scrubbing Deep Networks of Information Accessible from Input-Output Observations” European Conference on Computer Vision (2020)
>
> [7] Hayes, Jamie et al. “Inexact Unlearning Needs More Careful Evaluations to Avoid a False Sense of Privacy.” IEEE Conference on Secure and Trustworthy Machine Learning (2025)
>
> [8] Sun, Jialong et al. “Statistical MIA: Rethinking Membership Inference Attack for Reliable Unlearning Auditing.” arXiv preprint 2602.01150
>
> [9] Pawelczyk, Martin et al. "Machine Unlearning Fails to Remove Data Poisoning Attacks.” International Conference on Learning Representations (2025)
>
> [10] Wortsman, Mitchell, et al. "Robust fine-tuning of zero-shot models." Proceedings of the IEEE/CVF conference on computer vision and pattern recognition. 2022.
>
> [11] Siddiqui , Shoaib Ahmed et al. “From Dormant to Deleted: Tamper-Resistant Unlearning Through Weight-Space Regularization.” arxiv preprint 2505.22310

---

### Review · Reviewer_TdzB · 2026-06-30

**Summary Of Contributions:**

This paper proposes Minimally Invasive Machine Unlearning (MIMU), a machine unlearning framework that learns a control module to produce sparse parameter masks for a given forget request. The control module is instantiated as a GCN over a graph of parameters in a selected layer, where node features include local activations, parameter values, and gradients. At unlearning time, the learned controller is used to select a small set of parameters to mask, aiming to reduce predictability on the forget set while preserving retain-set behavior. The paper evaluates MIMU on MNIST, FMNIST, CIFAR-10, and ImageNet100, comparing against several unlearning baselines including Random Labels, Catastrophic Forgetting, Exact Unlearning, NegGrad+, L1-Sparse, SalUn, SSD, SCRUB, and retraining. The empirical results suggest that MIMU is computationally efficient per request and can achieve a strong tradeoff between class-level forgetting and retain-set accuracy.

I find the direction interesting. In particular, I like the idea of using a controller that respects the structure of neural-network weights through a graph representation, rather than treating all weights as independent scalar scores. The paper also runs a reasonably broad set of baselines and datasets, and the runtime advantage over optimization-based methods is potentially valuable for practical unlearning settings. However, several central aspects of the method are currently unclear or overclaimed. In particular, the mathematical formulation around the mask objective appears inconsistent with the stated unlearning goal, the top-$k$ / top-$\delta$ training procedure is under-specified, and the paper does not isolate the benefit of the GCN architecture against simpler learned-mask alternatives such as a per-weight MLP trained with the same objective.

**Audience:**

Yes

**Audience Explanation:**

I think the paper is relevant to the TMLR audience. Efficient approximate machine unlearning is an important topic, and the idea of amortizing unlearning through a learned controller is interesting. I also find the GCN-based weight-graph perspective promising, since it tries to exploit the structure of neural-network parameters rather than scoring weights independently.

That said, I view the novelty as moderate rather than high. Learning masks over neural-network weights is a well-established idea in pruning, continual learning, and related parameter-efficient adaptation settings. The main contribution here appears to be applying a learned parameter-graph controller to machine unlearning and showing a favorable per-request efficiency tradeoff. This is still potentially useful, especially under TMLR’s criteria, but the paper should more clearly position itself relative to learned masking/pruning methods and should include stronger ablations to isolate what the GCN contributes beyond simpler learned scoring functions.

**Broader Impact Concerns:**

The paper includes an impact statement and correctly notes that machine unlearning can be useful for privacy, safety, and model repair, but can also be misused for censorship or selective suppression. I do not see additional broader-impact concerns that would prevent acceptance. The main issue is that the paper should avoid overstating the privacy or regulatory implications of approximate unlearning unless it provides stronger evidence that the method satisfies the relevant unlearning definition.

**Claims And Evidence:**

Yes

**Claims Explanation:**

**Mostly yes, but important clarifications and revisions are needed.**

The empirical results support the claim that MIMU can be effective and fast for class-level unlearning. In Table 1, MIMU achieves near-zero forget accuracy while preserving retain accuracy reasonably well across the evaluated datasets. The runtime results also support the practical claim that MIMU has much lower per-request cost than several optimization-heavy baselines, especially on CIFAR-10 and ImageNet100. The entropy and saliency-map analyses are useful additional evidence toward the “minimally invasive” motivation.

However, I have several concerns about whether the current manuscript clearly and convincingly supports the broader claims.

First, the formal objective appears inconsistent. Equation 1 requires the masked model to preserve $P_\theta(y_i \mid x_i)$ on the forget set, which is the opposite of the stated goal of unlearning. Such a constraint would make sense for the retain set, but for the forget set one would expect the probability of the original label under the unlearned model to become small, or the predictive distribution to become high-entropy. The subsequent posterior-control derivation has the same issue: Equation 3 and Equation 4 optimize masks that preserve or explain the prediction under $m \odot \theta$, rather than directly optimizing the unlearned model, which would presumably be $(1-m)\odot\theta$ under the convention that selected weights are removed.

A charitable interpretation is that the controller is trained to identify a sparse predictive support subnetwork, which is then removed at unlearning time. This may be a reasonable heuristic. However, the paper does not clearly distinguish “finding parameters sufficient for prediction” from “removing parameters necessary for prediction.” In overparameterized networks, a sparse subnetwork can be sufficient without being necessary, because redundant pathways may support the same prediction. Thus, the current derivation does not fully justify why the learned mask should cause forgetting when removed. This should be clarified and the claims should be adjusted accordingly.

Second, the top-$\delta$ / top-$\kappa$ masking procedure is central but under-specified. The paper says it avoids optimization duality by “top-$\delta$ sampling” and later uses top-$\kappa$ filtering, but it is not fully clear how the top set is ranked, whether $\delta$ and $\kappa$ are cardinalities or fractions, whether the masks are hard or soft during training, whether a straight-through estimator is used, how the Gumbel-Top-$k$ relaxation is implemented, and whether the final deployment mask is binary. This matters because the controller is trained using its own current top-$k$ selections, which may introduce substantial selection bias or premature commitment: parameters outside the current top-$k$ set may receive little training signal even if they would form better masks.

Third, the “without inference-time optimization” claim should be made more precise. I agree that MIMU does not appear to run iterative fine-tuning or per-request mask optimization in the same way as many baselines. However, the controller input includes sample-specific activations and gradients, and Figure 1 explicitly shows forward and backward propagation at inference time. Therefore, the method is not simply a pure feed-forward map from a forget set to a mask. The paper should clarify whether it means “no iterative inference-time optimization,” “no weight update,” or “no backward pass.” These are different practical claims.

Fourth, the subclass/random unlearning results are much weaker than the class-level results. In Table 2, MIMU preserves retain accuracy but leaves high forget accuracy on several datasets. This is understandable for hard sparse masking, since fine-grained forget-set features are likely superposed with retain-set features, but it means the method is much less convincing as a general-purpose unlearning approach. The authors should discuss this limitation more directly and temper broad claims accordingly.

**Requested Changes:**

**Critical to my recommendation**

1. Fix and clarify the formal objective.

Equation 1 currently appears to preserve the forget-set prediction rather than remove it. The paper should clearly define the mask convention: does $m_k=1$ mean “keep” or “remove”? It should then distinguish the model used for support identification, e.g. $m\odot\theta$, from the unlearned model, e.g. $(1-m)\odot\theta$. If the intended method is “find a sparse predictive support and then remove it,” this should be stated explicitly, and the paper should discuss the missing assumption that sufficient predictive support is also necessary for the prediction.

2. Specify the top-$\delta$ / top-$\kappa$ procedure in reproducible detail.

The authors should clearly state how parameters are ranked, whether $\delta$ and $\kappa$ are absolute cardinalities or percentages, whether masks are hard or soft during training, whether the forward pass uses hard top-$k$ or soft relaxation, what Gumbel/Top-$k$ estimator is used, and whether/how temperature is annealed. The final unlearning-time mask should also be clearly specified.

3. Clarify the parameter graph construction, including biases.

The method is not currently reproducible because the graph construction is only described for weight parameters. The paper states that each graph node corresponds to an individual weight parameter and that edges connect parameters associated with the same neuron, but it does not specify how bias parameters are handled. Are biases included as graph nodes, ignored, always kept, or masked separately? If included, what are their node features and edges? This is a major implementation detail, especially because the method focuses on late layers where biases can directly affect class logits. The authors should provide a precise graph construction for dense and convolutional layers, including biases, and specify exactly which parameters are eligible for masking.

4. Add a simple learned-mask baseline, preferably an MLP trained with the same objective.

Since each weight already has a feature vector containing activations, weight value, and gradient, a natural baseline is a per-weight MLP that predicts the mask score independently, or an MLP with simple row/column/neuron summary features. This would test whether message passing over the proposed parameter graph is actually needed. The paper should report both performance and cost for GCN vs MLP so that the community can understand whether the graph structure is worth the additional complexity.

5. Clarify inference-time computation and report amortized cost.

The paper should distinguish offline controller-training cost, per-request feature extraction cost, GCN inference cost, and any filtering/search cost for $\kappa$. If gradients are computed for each forget request, this should be explicitly included in the runtime description. The authors should also report amortized cost as a function of the number of unlearning requests and the number of forget/representative samples.

6. Temper claims based on subclass/random unlearning.

The class-level results are strong, but Table 2 suggests that MIMU is much less effective for fine-grained subclass/random unlearning. The authors should discuss this as a limitation of sparse hard parameter masking, especially under feature superposition, rather than presenting the method as uniformly effective across unlearning regimes.

**Changes that would strengthen the work**

1. Quantify saliency-map preservation.

The saliency plots are visually useful, but the paper should include a quantitative metric for saliency shift on retain and forget samples. This would make the “prediction pattern preservation” claim stronger.

2. Ablate key input features to the controller.

Since the GCN input includes activations, parameter values, and gradients, it would be useful to ablate these features. In particular, the gradient feature seems central to the per-request information content.

3. Discuss soft masking as future work.

I do not think soft masking is necessary for this paper, but the authors may want to discuss it as a natural extension. Hard binary masking may be too coarse for subclass-level unlearning when features are superposed across retain and forget samples.

4. Improve writing and notation.

The paper would benefit from a careful pass to improve grammar, notation consistency, and clarity. In particular, the terms “masking,” “removing,” “control policy,” “top-$\delta$,” “top-$\kappa$,” and “representative samples” should be defined consistently.

---

> ### Author Response · Authors · 2026-07-14
>
> Thank you for reviewing our paper.
>
> We are pleased that you found the learned weight controller to be interesting and the weight-graph method promising. We also appreciate your comment that efficient approximate unlearning is an important topic. Below, we address the questions and concerns you raised in your review of our paper.
>
> **Fix and clarify the formal objective.**
>
> We appreciate you raising this. This is a central point of the paper, and we want to illuminate its precision. We have made edits to the introduction of section 3 to keep the intention of the paper clear.
> In our convention, $m_k = 1$ means that parameter $\\theta_k$ is selected as part of the predictive support for the forget set, while $m_k = 0$ means that the parameter is not selected. The control module is first trained to identify a sparse subset of parameters that is sufficient to preserve the original model's predictions on the forget set. Specifically, the masked support model
>
> $$f_{\\mathbf{m}_{\\text{forget}} \\odot \\theta}$$
>
> should remain closely aligned with the original model $$f_\\theta$$ on the forget samples:
>
> $$\left| P_\theta(y_i \mid x_i) - P_{\mathbf{m}{\text{forget}} \odot \theta}(y_i \mid x_i) \right| \leq \epsilon, \quad \forall (x_i, y_i) \in D{\text{forget}}$$
>
> subject to the sparsity constraint
>
> $$\\left\\| \\mathbf{m}_{\\text{forget}} \\right\\|_0 = \\delta$$
>
> where $\\delta$ is a predefined sparsity target and $\\odot$ denotes element-wise multiplication. This is not yet the unlearning stage. Instead, the control module identifies a set of $\\delta$ parameters that can still reproduce the prediction behavior of the original model.
> Afterwards, MIMU unlearns by removing those selected parameters from the model. The unlearned model is therefore defined as
>
> $$f_{\\theta_{\\text{unlearn}}}, \\qquad \\theta_{\\text{unlearn}} = (1 - \\mathbf{m}_{\\text{forget}}) \\odot \\theta$$
>
> This second stage masks out the parameters that were found to be most responsible for assigning the original class labels to forget-set samples, thereby reducing the model's predictive capacity on those samples. In this sense, MIMU finds a sparse predictive support, then removes it.
> This assumes that the sparse support identified by
>
> $$\\mathbf{m}_{\\text{forget}}$$
>
> reproduces the forget-set predictions and maintains them. If the model contains redundant alternative parameter pathways that all support the same predictions, then removing $\\mathbf{m}_{\\text{forget}} \\odot \\theta$ may reduce, but not fully eliminate, the model's predictive capacity on the forget set. However, as written in section 3.3, the final unlearning mask is a union of many sparse supports. And in our experimental results we showed that the model achieves near zero accuracy on the forget set while preserving the predictability of the model on the remaining data with these collected supports.
>
> **Specify the top-$\\kappa$ / top-$\\delta$ procedure in reproducible detail.**
>
> We have addressed your concerns in Appendix F and within text section 3.1.
> We follow the Gumbel subset sampling from the work of Xie and Ermon [1]. The Gumbel softmax subset sampling ranks the parameters by the magnitude of the output logit of the GCN along with noise from a Gumbel distribution. The parameters in the top percentage of magnitude after the noise addition are selected for the mask. The top-$\\delta$ is the percentage that determines the hard threshold of which parameters are selected during training. These parameters become ones in the hard (binary) mask output of the GCN. Gradients for training are maintained through the Gumbel Straight-Through Estimator [2]. Throughout training, the temperature of the Gumbel sampling process is kept constant and not annealed.
>
> Below is the PyTorch code for the Gumbel subset sampler implementation with Straight-Through Estimator:
> ```python
> top_k = math.ceil(len(logits) * k)
> gumbel_noise = -torch.log(-torch.log(torch.rand_like(logits) + eps) + eps)
> gumbel_logits = (logits + gumbel_noise) / temperature
> soft_mask = F.softmax(gumbel_logits, dim=-1)
> topK_indices = gumbel_logits.topk(k=top_k).indices
> hard_mask = torch.zeros_like(gumbel_logits, dtype=torch.float)
> hard_mask[topK_indices] = 1.0
> final_mask = (hard_mask - soft_mask).detach() + soft_mask
> ```
>
> The top-$\\kappa$ is the percentage that determines the number of parameters selected during inference time. As the GCN is already trained, Gumbel sampling is unnecessary. The top-$\\kappa$ percent of output are selected as the ones in the hard mask output of the GCN.
>
> The top-$\\delta$ and top-$\\kappa$ values are given as hyperparameters and are selected via ablation study as shown in Figures 5, 6, 12, and 13.

---

> > ### Author Response · Authors · 2026-07-14
> >
> > **Clarify the parameter graph construction, including biases.**
> >
> > We have revised the manuscript to make the implementation details of the GCN controller, as well as the complete training configuration and hyperparameters for each evaluated model, clearer. The GCN architecture and the model-specific training hyperparameters are described in Appendix D, Tables 7 and 8 accordingly. These tables provide all necessary details to reproduce the GCN training process for each experimental setting.
> >
> > **Add a simple learned-mask baseline, preferably an MLP trained with the same objective.**
> >
> > As per your request, we added Table 10 to the appendix, reproduced below for convenience. The GCN control module was replaced with an MLP with three layers. It was trained with the same activation, weight, and gradient inputs as the GCN. Table 11 shows the results across all datasets; the accuracy scores for FMNIST and ImageNet-100 are reproduced below.
> >
> > | Control Method | FMNIST Forget | FMNIST Retain | IN100 Forget | IN100 Retain |
> > |---|---|---|---|---|
> > | GCN | 0.73 ± 0.52 | 92.55 ± 0.46 | 0.00 ± 0.00 | 90.72 ± 0.08 |
> > | MLP | 80.01 ± 0.01 | 89.93 ± 1.38 | 0.00 ± 0.00 | 90.21 ± 0.12 |
> >
> > We observe that the MLP has worse retain set performance for all datasets. GCN is the better choice for preserving minimal invasion. GCN maintains the best overall performance.
> >
> > **Clarify inference-time computation and report amortized cost.**
> >
> > Thank you for mentioning this. We've added a runtime decomposition explanation in Appendix E that distinguishes the one-time offline training cost from the per-request inference cost and elaborates on each inference-time component.
> >
> > While MIMU does not perform any optimization during inference, each unlearning request does require one forward and backward pass through the original vision model to extract the gradients needed as input features to the GCN.
> >
> > For each representative sample, the GCN performs a single forward pass to generate the unlearning mask. The number of parameter indices, $\\kappa$, is specified by the user before unlearning. There is no filtering or search for $\\kappa$ performed at inference time. The number of forget-set representatives, $N_r^f$, is also given in advance of inference. MIMU only requires representative samples from the forget set, and no representatives are needed from the retain set. Finally, as mentioned in section 3.3, the final mask is the union of each mask from the representative samples.
> >
> > To address the request for amortized cost, we also include the following expressions in Appendix E:
> >
> > $$T_{\\text{amortized}} = \\frac{T_{\\text{train}} + R\\, T_{\\text{req}}}{R} = \\frac{T_{\\text{train}}}{R} + T_{\\text{req}}$$
> >
> > where $T_{\\text{train}}$ is the one-time offline controller-training cost (including parameter-graph construction, feature computation, and GCN training), $R$ is the number of unlearning requests, and $T_{\\text{req}}$ is the cost of a single request:
> >
> > $$T_{\\text{req}} = N_r^f \\left( T_{\\text{feat}} + T_{\\text{GCN}} \\right) + T_{\\text{apply}}$$
> >
> > **Temper claims based on subclass/random unlearning.**
> >
> > We have amended section 4.1 to address this concern regarding MIMU's performance for subclass unlearning. We would also like to clarify that this limitation is not unique to MIMU or sparse hard parameter masking. Studies evaluating random and subclass forgetting identify a trade-off between forgetting effectiveness and retained-model fidelity [3, 4]. More broadly, unlearning becomes harder when forgotten and retained examples are entangled in representation space, with lower forgetting scores across multiple algorithms [5].
> > Hard masking may be poorly aligned with the localized feature changes needed for MIMU's fine-grained settings, especially under feature superposition [6], but the underlying limitation is widespread across current unlearning methods.

---

> > > ### Author Response · Authors · 2026-07-14
> > >
> > > **Quantify saliency-map preservation.**
> > >
> > > The details of saliency-map construction and computation are described in detail in Appendix C, and the corresponding quantitative results for the retain and forget sets are reported in Table 5.
> > >
> > > **Ablate key input features to the controller.**
> > >
> > > As requested, we conducted an ablation study that shows the impact of the different inputs to the GCN. Table 10 shows the complete results, but the accuracy scores for FMNIST and ImageNet-100 are reproduced below.
> > > | GCN Ablation | FMNIST Forget | FMNIST Retain | IN100 Forget | IN100 Retain |
> > > |---|---|---|---|---|
> > > | Original | 0.73 ± 0.52 | 92.55 ± 0.46 | 0.00 ± 0.00 | 90.72 ± 0.08 |
> > > | No In Activations | 20.31 ± 3.45 | 92.91 ± 0.57 | 0.00 ± 0.00 | 89.88 ± 0.08 |
> > > | No Out Activations | 0.01 ± 0.01 | 84.93 ± 2.17 | 0.00 ± 0.00 | 89.94 ± 0.19 |
> > > | No Weights | 42.47 ± 8.76 | 92.65 ± 0.65 | 57.40 ± 2.52 | 91.47 ± 0.02 |
> > > | No Gradients | 71.72 ± 6.86 | 86.85 ± 1.90 | 3.63 ± 1.27 | 99.65 ± 0.02 |
> > >
> > > **Improve writing and notation.**
> > >
> > > Thank you for the suggestion. We've carefully gone through the paper and have cleaned up the writing to keep the intent of the paper crystal clear for future readers. All changes have been highlighted in red in the revised manuscript.
> > >
> > > [1] Xie, Sang Michael & Ermon, Stefano. "Reparameterizable Subset Sampling via Continuous Relaxations." *Proceedings of the Twenty-Eighth International Joint Conference on Artificial Intelligence* (2019)
> > >
> > > [2] Jang, Eric et al. "Categorical Reparameterization with Gumbel-Softmax." *Proceedings of the International Conference on Learning Representations* (2017)
> > >
> > > [3] Lin, et al. "GDR-GMA: Machine Unlearning via Direction-Rectified and Magnitude-Adjusted Gradients." *MM '24: Proceedings of the 32nd ACM International Conference on Multimedia* (2024)
> > >
> > > [4] Mavrothalassitis, Ioannis et al. "Ascent Fails to Forget." *Advances in Neural Information Processing Systems* (2025)
> > >
> > > [5] Zhao, et al. "What Makes Unlearning Hard and What to Do About It." *Advances in Neural Information Processing Systems* (2024)
> > >
> > > [6] Elhage, et al. "Toy Models of Superposition." *Transformer Circuits Thread* (2022)

---

> > > > ### Comment · Reviewer_TdzB · 2026-07-18
> > > >
> > > > Thank you for the substantial revision. The added Gumbel Top-$k$ details, feature ablations, saliency quantification, runtime decomposition, and GCN-versus-MLP comparison address most of my concerns. I now find the paper’s narrower story convincing: MIMU is an efficient amortized method for predictive suppression, especially for class-level unlearning, rather than a guarantee of removing all training-data influence. I am leaning toward acceptance.
> > > >
> > > > A few issues still need correction or clarification:
> > > >
> > > > 1. **Page 3 mask expression.** The unlearned model should be written as
> > > >    $$
> > > >    \tilde{\theta}=(1-m_{\mathrm{forget}})\odot\theta,
> > > >    $$
> > > >    not $1-m_{\mathrm{forget}}\odot\theta$. The parentheses are essential.
> > > >
> > > > 2. **Bias parameters remain unspecified.** Table 7 describes biases in the GCN controller, not biases in the target model. Please state whether target-model biases are masked, always retained, absent, or represented separately. If included, define their node features and graph connections.
> > > >
> > > > 3. **“Via forward propagation” is misleading.** Appendix E states that each request requires a forward and backward pass through the original model, followed by a GCN forward pass. The accurate claim is “without iterative request-time optimization” or “without request-time parameter updates.”
> > > >
> > > > 4. **MIA ROC-AUC values below 0.5 require care.** An AUC of (0.13) may indicate strong separability in the reverse direction, not attack failure. Please report a direction-invariant measure such as $\max(\mathrm{AUC},1-\mathrm{AUC})$, reverse the score when appropriate, or justify why inversion is not allowed.
> > > >
> > > > 5. **The Gumbel Top-$k$ ranking description is inconsistent with the code.** The rebuttal/text says parameters are ranked by the **magnitude** of the GCN output logits, but the provided implementation applies `topk` directly to `gumbel_logits`, not to `abs(gumbel_logits)`. Please clarify which is intended and make the prose and implementation consistent. Relatedly, please state whether the gradient node feature is signed, absolute-valued, or otherwise normalized.
> > > >
> > > > These points do not change my positive overall assessment. Subject to these revisions, I am leaning accept.

---

### Review · Reviewer_9z2M · 2026-07-01

**Summary Of Contributions:**

The paper proposes an approach to unlearn data points in a forget set by way of a control module that maps unlearning requests to a weight sparsity mask. The method, dubbed minimally invasive machine unlearning, constructs this control module via an optimization criterion derived from a posterior control objective.

**Strengths:**
- the paper is, for the most part, well written (Section 2.1 is particularly useful in contextualizing the work)
- the idea is grounded in intuition and plausible, with results that support the argumentation
- for those unfamiliar in this space, the presentation of the paper yields a working mental framework with which to think about the problem(s)
- the analysis of predictive confidence is insightful and important in validating claims

**Weaknesses:**
- little discussion is paid to scaling, aside from comments about the last couple layers being important
- building from the above, it is unclear how this could scale to modern neural networks; the evaluated models and datasets are outdated


Minor presentation issues:
- Citation commands are used incorrectly. For example, “...and approximate unlearning Neel et al. (2021)". Here, a parenthetical citation would be appropriate, i.e., “...and approximate unlearning (Neel et al., 2021).”, since the citation is not integrated into the sentence narrative.
- Grammar breaks down in Section 3.3, e.g., "..can destroy prediction pattern of some retain data points" and the lone "Under" before Section 4 begins.

**Audience:**

Yes

**Audience Explanation:**

Efficient and effective machine unlearning is an important and open challenge. Although the authors focus on small CNNs, the findings could lend themselves useful in scaling to modern transformer architectures.

**Broader Impact Concerns:**

No concerns.

**Claims And Evidence:**

Yes

**Claims Explanation:**

Section 4.1 provides extensive baseline comparisons to give clear evidence when/where the proposed method, MIMU, outperforms existing baselines. This supports the initial claim that MIMU achieves competitive results against state-of-the-art unlearning baselines. The inference-time efficiency claims are supported by Figure 3 and Table 6.

The claims for unlearning vs. high confidence mispredictions are supported by Table 3.

**Requested Changes:**

1. Please fix the citation and grammar issues mentioned above.
2. Can you discuss how this could scale to modern neural architectures, e.g., vision transformers?

In the contributions, you write:
> it drives predictive confidence on the forget set toward high predictive entropy, rather than confidently misclassifying forgotten samples into incorrect classes...

3. In section 2.1, it would be great to add a comment about why this delineation is an important part of the problem. I view that clarification as a missing connection in that section.

---

> ### Author Response · Authors · 2026-07-14
>
> Thank you for reviewing our paper. We are glad that you found the paper well written and that you appreciated the intuition, empirical grounding, and analysis of predictive confidence. We also appreciate your comment that the presentation provides a useful mental framework for readers who may be less familiar with this area.
>
> We have carefully addressed your suggestions and concerns in the revised manuscript. In particular, we have expanded the discussion on scalability (appendix Section I), clarified how MIMU can be applied beyond the evaluated settings, and added further explanation regarding its applicability to larger neural networks.
>
> We have also addressed the minor presentation issues you identified. Specifically, we corrected improper citation usages throughout the paper, including cases where parenthetical citations were more appropriate than narrative citations. In addition, we carefully revised Section 3.3 to fix grammar issues, including the phrasing around “prediction patterns of some retain data points,” and removed the stray word “Under” before Section 4. We also revised Section 2.1 to explain why this delineation is an important part of the problem, as suggested.
>
> These changes have been highlighted in red in the revised manuscript.

---

### Comment · Action_Editor_swNF · 2026-07-18

The paper proposes MIMU, an amortized machine-unlearning framework that uses a GCN-based controller to generate sparse parameter masks for forget requests. The method aims to suppress predictions on forgotten data while preserving retained behavior, without iterative request-time optimization. Reviewers generally found the idea interesting and the class-level results and runtime advantages promising. In response, the authors clarified the mask convention and Gumbel top-K procedure, added controller and feature ablations, quantified saliency changes, discussed amortized cost and scalability, and included membership-inference results.

The discussion should now focus on whether the revisions resolve the central concerns. In particular, reviewers are invited to assess whether identifying a sparse subnetwork sufficient for a prediction adequately justifies forgetting after its removal, and whether the paper now clearly limits its claims to predictive suppression rather than removal of training influence. Further comments on fine-grained and sequential unlearning, recovery or relearning behavior, scalability beyond the tested models, and the interpretation of ROC-AUC values below 0.5 would be especially helpful.